

# Does high-latitude ionospheric electrodynamics exhibit hemispheric mirror symmetry?

Spencer Mark Hatch[1], Heikki Vanhamäki[2], Karl Magnus Laundal[1], Jone Peter Reistad[1], Johnathan
K. Burchill[3], Levan Lomidze[3], David J. Knudsen[3], Michael Madelaire[1], and Habtamu Tesfaw[2]

[1]Department of Physics and Technology, University of Bergen, Bergen, Norway
[2]Space Physics and Astronomy Research Unit, University of Oulu, Oulu, Finland
[3]Department of Physics and Astronomy, University of Calgary, Calgary, Alberta, Canada
**Correspondence:** Spencer Mark Hatch (spencer.hatch@uib.no)

**Abstract.** Ionospheric electrodynamics is a problem of mechanical stress balance mediated by electromagnetic forces. Joule heating (the total rate of frictional heating of thermospheric gases and ionospheric plasma) and ionospheric Hall and Pedersen conductances comprise three of the most basic descriptors of this problem. More than half a century after identification of their central role in ionospheric electrodynamics several important questions about these quantities, including the degree to

which they exhibit hemispheric symmetry under reversal of the sign of dipole tilt and the sign of the $y$ component of the interplanetary magnetic field (so-called "mirror symmetry"), remain unanswered. While global estimates of these key parameters can be obtained by combining existing empirical models, one often encounters some frustrating sources of uncertainty: the measurements from which such models are derived, usually magnetic field and electric field or ion drift measurements, are typically measured separately and do not necessarily align. The models to be combined moreover often use different input pa-

rameters, different assumptions about hemispheric symmetry, and/or different coordinate systems. We eliminate these sources of uncertainty in model predictions of electromagnetic work $\mathbf{J} \cdot \mathbf{E}$ (in general not equal to Joule heating $\eta J^2$) and ionospheric conductances by combining two new empirical models of the high-latitude ionospheric electric potential and ionospheric currents that are derived in a mutually consistent fashion: these models do not assume any form of symmetry between the two hemispheres; are based on Apex coordinates, spherical harmonics, and the same model input parameters; and are derived ex-

clusively from convection and magnetic field measurements made by the Swarm and CHAMP satellites. The model source code is open source and publicly available. Comparison of high-latitude distributions of electromagnetic work in each hemisphere as functions of dipole tilt and interplanetary magnetic field clock angle indicate that the typical assumption of mirror symmetry is largely justified. Model predictions of ionospheric Hall and Pedersen conductances exhibit a degree of symmetry, but clearly asymmetric responses to dipole tilt and solar wind driving conditions are also identified. The distinction between

electromagnetic work and Joule heating allows us to identify where and under what conditions the assumption that the neutral wind corotates with the earth is not likely to be physically consistent with predicted Hall and Pedersen conductances.



## 1   Introduction

At high latitudes, the Earth's ionosphere is electrodynamically coupled to the magnetosphere and the solar wind via the Earth's magnetic field, and mechanically coupled to the neutral atmosphere via collisions. When the interplanetary magnetic field (IMF) carried by the solar wind points southward, for example, the IMF reconnects with Earth's magnetic field lines on the dayside and drags these reconnected field lines over the Earth's polar cap to the nightside. These field lines reconnect in the magnetotail and circulate back to the dayside.

Because collisions between charged particles are very infrequent in the magnetosphere, magnetospheric plasma is frozen to the Earth's field lines as it undergoes convection. In contrast, the overlapping ionosphere-thermosphere region at lower altitudes is highly collisional. Here, ionospheric plasma is dragged through the neutral thermospheric gas at speeds of several hundred meters to several kilometers per second, resulting in energy dissipation via plasma-neutral friction that can reach nearly a terawatt globally (Billett et al., 2018). This frictional heating, which is often referred to as "Joule dissipation" or "Joule heating", represents one of the most important processes by which energy is transferred from the solar wind to the ionosphere-thermosphere system.

Heelis and Maute (2020), Sarris (2019), and Richmond (2021) have all recently pointed out aspects of energy transfer from the solar wind via Joule heating that remain poorly understood. One of the most significant of these is ionospheric conductivity, which is central to understanding magnetosphere-ionosphere coupling and is a required input for many empirical and numerical models but is, as Weimer and Edwards (2021) have stated, arguably one of the least measured and estimated parameters. More generally, there are overall far fewer estimates of quantities that are central in describing MIT coupling in the Southern Hemisphere (SH) relative to the Northern Hemisphere (NH). One is therefore often left to assume that a quantity measured in the SH is the same as the quantity measured in the NH at a magnetically conjugate point when the signs of the $y$ component of the IMF ($B_y$) and the tilt angle of the Earth's dipole $\Psi$ are flipped. This assumption of hemispheric symmetry is typically formulated

$$Q_{\mathrm{NH}}(\mathrm{MLat}, \Psi, B_y, \dots) \pm Q_{\mathrm{SH}}(-\mathrm{MLat}, -\Psi, -B_y, \dots) = 0, \tag{1}$$

where $Q$ is a quantity such as Joule heating at conjugate points in the two hemispheres, and the choice of sign depends on which quantity is being considered.

This assumption has played a major role in global empirical models of high-latitude ionospheric convection: Since the 1980s at least 15 such empirical models have been created (see, e.g., review in Cousins and Shepherd, 2010), and the major data sets from which these models are derived lack comprehensive observations in one (typically the Southern) or both hemispheres. Hence many of these models assume some form of hemispheric symmetry of necessity, even though differences in ionospheric convection and current patterns exist between the two hemispheres (Cousins and Shepherd, 2010; Förster and Haaland, 2015; Hatch et al., 2022).

One problem with the assumption of hemispheric symmetry is that it obscures other sources of uncertainty. For example, Weimer and Edwards (2021) used three separate empirical models that all make different assumptions about hemispheric symmetry to estimate critical ionospheric parameters such as Pedersen conductance $\Sigma_P$, Hall conductance $\Sigma_H$, Joule heating,





and perturbation Poynting flux $\mathbf{S}_p$. They report that the resulting estimates of $\Sigma_P$ and $\Sigma_H$ are in some places unphysical (too high or negative). While they were not able to determine the source of these unphysical estimates, possible sources of error include the assumption of hemispheric symmetry and the combination of different empirical models that are not necessarily derived in a mutually consistent fashion. By "mutually consistent derivation" we mean that the coordinate systems, model parameters and physical assumptions are, as much as possible, the same in the derivation of each model.

One of the purposes of this study is to derive an appropriate set of empirical models for high-latitude ionospheric electrodynamics that treats the two hemispheres independently but equally and that are derived in a mutually consistent fashion. The primary challenge for such a set of models is a comprehensive set of observations in both hemispheres. The magnetic field measurements and recently released multi-year ion drift measurements made by the Swarm satellites in each hemisphere are appropriate for meeting this challenge. In this study we use these measurements to derive the first such set of mutually consistently derived empirical models.

In Section 2 we describe our approach and define the quantities that we aim to estimate. In Section 3 we outline the derivation of an empirical model, hereafter referred to as the Swarm High-latitude Convection ("Swarm Hi-C") model, of the high-latitude electric potential $\Phi$, the convection electric field $\mathbf{E} = -\nabla\Phi$, and the plasma convection

$$\mathbf{v}_E = \mathbf{E} \times \mathbf{B}_0 / B_0^2. \tag{2}$$

A central goal in creation of the Swarm Hi-C model is consistency with the Average Magnetic field and Polar current System (AMPS) model presented by Laundal et al. (2018). We also describe the Swarm measurements and other data sets that are used. In Section 4 we compare Swarm Hi-C model ionospheric electric potentials in the Northern and Southern Hemispheres for different IMF clock angles and dipole tilts, and compare Swarm Hi-C cross-polar cap potential (CPCP) values with CPCP values reported in previous convection studies. In Section 5 we combine outputs from the Swarm Hi-C model and the AMPS model (i.e., the Swarm Ionospheric Polar Electrodynamics, or Swipe, model; see Hatch, 2023) to estimate height-integrated electromagnetic work $\mathbf{J}_\perp \cdot \mathbf{E}_\perp$ and Hall and Pedersen conductances $\Sigma_H$ and $\Sigma_P$ at high latitudes. In Section 6 we discuss our findings. We then summarize and conclude.

## 2 Background

The goal of this study is to empirically estimate and make interhemispheric comparisons of three quantities that are central to describing MIT coupling: height-integrated electromagnetic work $W$ (which is not necessarily the same as the height-integrated Joule heating rate $W_J$ as we describe below) and height-integrated Hall and Pedersen conductivities (Hall and Pedersen conductances) $\Sigma_H$ and $\Sigma_P$.

Our starting point for deriving these quantities is the perpendicular component of the height-resolved ionospheric Ohm's law in steady state (that is, assuming steady-state stress balance between Lorentz and collisional drag forces and neglecting all other forces in the ion momentum equation; see Section 3.2.1 in Vasyliunas, 2012):

$$\mathbf{j}_\perp = \sigma_P \left( \mathbf{E}_\perp + \mathbf{v}_n \times \mathbf{B}_0 \right) + \sigma_H \mathbf{b} \times \left( \mathbf{E}_\perp + \mathbf{v}_n \times \mathbf{B}_0 \right), \tag{3}$$



with $\sigma_P$ and $\sigma_H$ Pedersen and Hall conductivities, $\mathbf{E}_\perp$ and $\mathbf{B}_0$ the ionospheric electric field and background geomagnetic field, and $\mathbf{v}_n$ the neutral wind. The corresponding local Joule heating rate is

$$w_J = \mathbf{j}_\perp \cdot (\mathbf{E}_\perp + \mathbf{v}_n \times \mathbf{B}_0) = \sigma_p |(\mathbf{E}_\perp + \mathbf{v}_n \times \mathbf{B}_0)|^2. \tag{4}$$

The definition of Joule heating is in some studies omitted or imprecise (see discussions in Vasyliunas and Song, 2005; Strangeway, 2012; Mannucci et al., 2022), and one occasionally encounters statements in the literature about "the neutral wind contribution to Joule heating" or "the effect of neutral winds on Joule heating". This language arises from the typically used expressions for height-resolved and height-integrated Joule heating, Equations 4 and 8, from which one may get the incorrect impression that the Joule heating rate is divided into separate contributions from the electric field and the neutral winds and magnetic field, respectively $\mathbf{j}_\perp \cdot \mathbf{E}_\perp$ (electromagnetic work) and $\mathbf{j}_\perp \cdot (\mathbf{v}_n \times \mathbf{B}_0) = -\mathbf{v}_n \cdot (\mathbf{j}_\perp \times \mathbf{B}_0)$. The latter is sometimes referred to as the "neutral wind dynamo term". While it is true that Joule heating in Equation 4 may be mathematically expressed as the sum of these terms,

$$w_J = \mathbf{j}_\perp \cdot \mathbf{E}_\perp + \mathbf{j}_\perp \cdot (\mathbf{v}_n \times \mathbf{B}_0), \tag{5}$$

one must nonetheless be aware that these two terms are frame dependent and in general can be negative or positive (see discussion in e.g. Matsuo and Richmond, 2008; Richmond, 2010; Cai et al., 2013). In contrast the LHS (Joule heating) represents the sum total of the heating rates of all neutral and plasma populations (Strangeway, 2012), and is therefore necessarily non-negative and frame-invariant under Galilean relativity (specifically "magnetic" Galilean relativity, see Mannucci et al., 2022). That this is true is seen in the expression given for Joule heating by Mannucci et al. (2022), their Equation 26:

$$w_J = n_e^2 \eta |\mathbf{B}_0 \times [m_i \nu_{in}(\mathbf{v}_i - \mathbf{v}_n) + m_e \nu_{en}(\mathbf{v}_e - \mathbf{v}_n)]|^2 / B_0^4, \tag{6}$$

where $\mathbf{v}_i$ and $\mathbf{v}_e$ are respectively the ion and electron drift velocities, $\nu_{in}$ and $\nu_{en}$ are the ion- and electron-neutral collision frequencies for momentum exchange, and $\eta$ is a resistivity. This expression for Joule heating is equivalent to Equation 4 but has the advantage of underscoring that (i) Joule heating is frame-invariant and non-negative, and (ii) it makes little sense to speak of a "neutral wind contribution" to Joule heating, since the definition of Joule heating is intrinsically tied to the reference frame of the neutrals.

Proceeding with the derivation, under the assumptions that

1. Magnetic field lines are approximately radial, such that $\mathbf{b} \approx \mp \mathbf{r}$;

2. The electric field $\mathbf{E}_\perp$ is independent of altitude over ionospheric $E$- and $F$-region altitudes ($\sim$100–250 km);

3. The neutral wind is constant over ionospheric $E$- and $F$-region altitudes in Earth's rotating frame of reference;

integration of Equations 3 and 4 over altitude yields

$$\mathbf{J}_\perp = \Sigma_P (\mathbf{E}_\perp + \mathbf{v}_n \times \mathbf{B}_0) + \Sigma_H \mathbf{b} \times (\mathbf{E}_\perp + \mathbf{v}_n \times \mathbf{B}_0); \tag{7}$$





$$W_J = \int w_J \, dr = \mathbf{J}_\perp \cdot (\mathbf{E}_\perp + \mathbf{v}_n \times \mathbf{B}_0). \tag{8}$$

The first assumption is a decent approximation at high latitudes in both hemispheres, though less so in the Southern Hemisphere where the geomagnetic field inclination and field strength vary more strongly with latitude and longitude (see Figure 2 in Laundal et al., 2017). The second assumption is likewise a decent approximation at altitudes over which the electron mobility $k_e = \Omega_e/\nu_{en}$ exceeds 1, generally true above the $D$ region. The third assumption is in general not justified because of the near-permanent presence of vertical shears in neutral wind altitudes profiles (Larsen, 2002), but is made here of necessity because otherwise a distinction between the average neutral wind weighted by Hall and Pedersen conductivity altitude profiles must be retained and addressed. While outside the scope of this study, investigating this topic is a future priority.

Taking the dot product and cross product of the height-integrated Ohm's law (7) with $\mathbf{E}_\perp$ one finds (Amm, 2001)

$$\Sigma_H = \mp \mathbf{r} \cdot \left[\mathbf{J}_\perp \times (\mathbf{E}_\perp + \mathbf{v}_n \times \mathbf{B}_0)\right]/|\mathbf{E}_\perp + \mathbf{v}_n \times \mathbf{B}_0|^2; \tag{9}$$

$$\Sigma_P = \mathbf{J}_\perp \cdot (\mathbf{E}_\perp + \mathbf{v}_n \times \mathbf{B}_0)/|\mathbf{E}_\perp + \mathbf{v}_n \times \mathbf{B}_0|^2; \tag{10}$$

where the upper and lower signs of the RHS in the expression for $\Sigma_H$ are respectively for the Northern and Southern Hemisphere.

Lacking an appropriate model of the neutral wind, to estimate these conductances we must assume that the contribution of the term $\mathbf{v}_n \times \mathbf{B}_0$ in Equations 9–10 is small compared to that of $\mathbf{E}_\perp$. We therefore estimate height-integrated electromagnetic work and ionospheric conductances via the expressions

$$W = \mathbf{J}_\perp \cdot \mathbf{E}_\perp; \tag{11}$$

$$\Sigma_H = \mp \mathbf{r} \cdot \left[\mathbf{J}_\perp \times \mathbf{E}_\perp\right]/|\mathbf{E}_\perp|^2; \tag{12}$$

$$\Sigma_P = \mathbf{J}_\perp \cdot \mathbf{E}_\perp/|\mathbf{E}_\perp|^2. \tag{13}$$

We do not refer to the height-integrated electromagnetic work $W$ in Equation 11 as an estimate of the height-integrated Joule heating $W_J$ since $W$ can be negative, whereas Joule heating as defined above is always positive. This distinction is not trivial, as it represents information that enables us to assess where our estimates of the Hall and Pedersen conductance may be valid, as we show in Sections 5.2 and 5.3.

Throughout this study the height-integrated perpendicular currents $\mathbf{J}_\perp = \nabla \times \Delta\mathbf{B}/\mu_0$ (where $\Delta\mathbf{B}$ is the perturbation magnetic field) and the convection electric field $\mathbf{E}_\perp = -\nabla\Phi$ are respectively calculated from the AMPS model (Appendix B in Laundal et al., 2018) and the Swarm High-latitude Convection, or "Swarm Hi-C", model (Section 3). Thus estimates of these two quantities, and consequently also height-integrated electromagnetic work $W = \mathbf{J}_\perp \cdot \mathbf{E}_\perp$, do not rely on Ohm's law and are unaffected by the third assumption listed above (i.e., neutral wind $\mathbf{v}_n$ does not vary with altitude).

One could also use the models we present to estimate the perturbation Poynting flux

$$\mathbf{S}_p = \frac{\mathbf{E}_\perp \times \Delta\mathbf{B}}{\mu_0}. \tag{14}$$





We plan to address this in a possible future study, but observe for completeness that the Poynting flux and height-integrated Joule heating are sometimes used almost interchangeably (e.g., Rastätter et al., 2016; Weimer, 2005, and references therein).

That these quantities do not always correspond in a point-by-point fashion has been shown using both synthetic data (Vanhamäki et al., 2012) and at least one set of empirical models (Weimer and Edwards, 2021). Richmond (2010) has pointed out the precise conditions under which the Poynting flux "may be used to estimate the field line-integrated electromagnetic energy dissipation" (roughly height-integrated Joule heating $W$). The neutral wind $\mathbf{v}_n$ notably does not appear in Equation 14, as the Poynting flux is frame dependent.

## 3 Methodology and data

The input data for the Swarm Hi-C model are Swarm TII measurements of the cross-track ion drift velocity $\mathbf{v}_i \cdot \hat{\mathbf{y}} = v_{i,y}$. The unit vector $\hat{\mathbf{y}}$ points along the $y$ component of the coordinate system defined by the satellite track: $\hat{\mathbf{x}}$ is in the direction of the satellite velocity, and $\hat{\mathbf{y}}$ is perpendicular to $\hat{\mathbf{x}}$ and horizontally to the right when facing the direction of motion. In the most recent release (version 0302) of the Swarm TII 2-Hz cross-track flow dataset, this is the quantity "Viy". We use available data

from 2014-05-01 (i.e., after the Swarm commissioning period) to 2023-04-15. We additionally apply the following constraints.

1. We only use measurements of $v_{i,y}$ that are flagged as calibrated, as indicated by the second bit of the quantity "Quality_flags" being set to 1 in v0302 of the TII cross-track flow dataset; see Burchill and Knudsen (2022) or section 3.4.1.1 in "EFI TII Cross-Track Flow Data Release Notes" (Burchill and Knudsen, 2020). (Lomidze et al., 2021, showed that statistical maps of high-latitude ion convection derived from v0302 of Swarm TII cross-track data are consistent with

corresponding estimates from the Weimer, 2005 model.)

2. We exclude measurements made equatorward of Quasi-Dipole latitudes $\pm 44°$, as this is the low-latitude boundary used for calibration (see Section 3.2).

3. For each individual 2-Hz TII NASA CDF file, we retain every tenth measurement such that the effective measurement cadence is 5 s, or approximately every 40 km.

Each $v_{i,y}$ measurement is associated with 1-min OMNI data that is time shifted to the bow shock, and averaged over the preceding 20 min. We have chosen this averaging window for the reasons given by Laundal et al. (2018): (i) the high-latitude pattern of currents and energy input take 10s of minutes to adapt to a recent change in driving conditions at the magnetopause (Snekvik et al., 2017; Billett et al., 2022; Pedersen et al., 2023); (ii) small-scale spatial variations and turbulence within the solar wind may render the instantaneously measured solar wind conditions an inappropriate indicator of the larger-scale solar wind

conditions; (iii) the time shift from the solar wind monitor to the magnetopause is not perfect. This choice has the additional advantage of being consistent with the treatment of solar wind and IMF measurements in the derivation of the AMPS model.

Figure 1 shows the distributions of solar wind and IMF conditions (top row); F10.7 and dipole tilt (middle row); and Swarm satellite magnetic latitude (MLat), magnetic local time (MLT), and altitude (bottom row). Unless otherwise specified, throughout this study magnetic coordinates are given in Modified Apex coordinates at a reference altitude $h_R = 110$ km



(hereafter denoted MA-110 coordinates), since in this coordinate system the magnetic latitude is constant along a given field line such that convection velocity and electric field may be mapped along field lines. In the top four panels the black lines indicate the weighted distribution using the Huber weights of the last model iteration (see Section 3.4 for more information about Huber weights). As stated by Laundal et al. (2018), "If the model representation was flawed for more extreme conditions, the Huber weighted distributions would be expected to more strongly peak at the most frequent conditions and go to zero at the

ends where the data fit would be poor." That this is not the case indicates that the Swarm Hi-C model generally gives a good average representation of the ionospheric convection.

Regarding the distribution of measurements, although MLT coverage is fairly uniform (bottom left panel in Figure 1), the Swarm satellite orbits are biased in their coverage of MA-110 magnetic longitudes (not shown; see Figure 11 in Hatch et al., 2022). The regions of highest measurement density ($>1.5$ measurements/km$^2$) in the NH are located between approximately

135° and 225° magnetic longitude (MLon), and in the SH between 0° and 45° MLon. This sampling issue is not likely to bias the model in the Northern Hemisphere since the field inclination is generally very high in the NH polar region, but it could be an issue for the SH measurements where the field inclination varies relatively much more with magnetic longitude and latitude. On the other hand, to the extent that this bias is directly and only related to local distortions of the geomagnetic field geometry, it is accounted for by our use of Apex coordinates.

Using MA-110 coordinates the electric field may be written

$$\mathbf{E}_\perp = E_{d1}\mathbf{d}_1 + E_{d2}\mathbf{d}_2, \tag{15}$$

with $\mathbf{d}_1$ and $\mathbf{d}_2$ non-orthogonal basis vectors that point, respectively, more or less in the magnetic eastward and equatorward directions. Then (Equations 4.8–4.9 in Richmond, 1995)

$$E_{d1} = -\frac{1}{(R_E + h_R)\cos\lambda}\frac{\partial\Phi}{\partial\phi};$$
$$E_{d2} = \frac{1}{(R_E + h_R)\sin I_m}\frac{\partial\Phi}{\partial\lambda}; \tag{16}$$

where $\Phi$ is the electric potential, the radius is $R_E + h_R$, with $R_E = 6371.2$ km the radius of Earth, $\phi$ is the magnetic local time (MLT) in degrees (e.g., 1 h MLT = 15°), and $\lambda$ is MA-110 latitude. The quantity $\sin I_m = 2\sin\lambda\left(4 - 3\cos^2\lambda\right)^{-1/2}$, with $I_m$ the field inclination, and $\cos\lambda = \sqrt{R/(R_E + h_A)}$ with $h_A$ the apex altitude of the field line in question.

Similar to $\mathbf{E}_\perp$, the convection velocity

$$\mathbf{v}_E \equiv \mathbf{E} \times \mathbf{B}_0/B_0^2 = v_{e1}\mathbf{e}_1 + v_{e2}\mathbf{e}_2, \tag{17}$$

with

$$v_{e1} = E_{d2}/B_{e3};$$
$$v_{e2} = -E_{d1}/B_{e3};$$
$$B_{e3} = B_0/D;$$
$$D \equiv |\mathbf{d}_1 \times \mathbf{d}_2|; \tag{18}$$





**Figure 1.** Data distribution for ∼19 million Swarm TII cross-track convection velocity measurements in same format as Figure 1 in Laundal et al. (2018). The black lines indicate the weighted distribution using the Huber weights of the last model iteration.





and $\mathbf{e}_1$ and $\mathbf{e}_2$ non-orthogonal vectors that respectively approximately point in the magnetic eastward and equatorward directions. Laundal et al. (2018) use CHAOS-6 to get $\mathbf{B}_0$ whereas we use the value of the magnetic field provided in the Swarm TII cross-track flow dataset, which is the magnetic field measured by Swarm at 1 Hz upsampled by interpolation to 2 Hz.

We wish to use Equation 17 together with Swarm EFI ion drift measurements to model ionospheric convection at 110-km altitude, the reference height of MA-110 coordinates. When only one component of $\mathbf{v}_E$ is measured along a unit vector $\hat{\mathbf{y}}$, we have from Equation 17 (Equation 8.2 in Richmond, 1995)

$$\hat{\mathbf{y}} \cdot \mathbf{v}_E = \frac{E_{d2}}{B_{e3}} \hat{\mathbf{y}} \cdot \mathbf{e}_1 - \frac{E_{d1}}{B_{e3}} \hat{\mathbf{y}} \cdot \mathbf{e}_2. \tag{19}$$

Care must be taken in relating this expression to Swarm EFI measurements since the ion drift measured along EFI instrument's
$y$ axis, $\mathbf{v}_{i,y} = v_{i,y}\hat{\mathbf{y}}$, can and often does include a (typically small) component along $\mathbf{B}_0$, whereas the convection velocity $\mathbf{v}_E$ in Equation 17 has no component along $\mathbf{B}_0$. (The magnitude of $|\hat{\mathbf{y}} \cdot \hat{\mathbf{b}}_0|$ is 0.07 or less for 50% of all measurements, and 0.26 or less for 90% of all measurements.) To address this we define a new unit vector $\hat{\mathbf{y}}_\perp$ that does not have a component along $\hat{\mathbf{b}}_0$ (the unit vector pointing in the direction of $\mathbf{B}_0$):

$$\mathbf{y}_\perp = \hat{\mathbf{y}} - \left(\hat{\mathbf{y}} \cdot \hat{\mathbf{b}}_0\right)\hat{\mathbf{b}}_0;$$

$$\hat{\mathbf{y}}_\perp = \mathbf{y}_\perp / |\mathbf{y}_\perp|. \tag{20}$$

We may then make the identification $\hat{\mathbf{y}} \cdot \mathbf{v}_E = \mathbf{v}_{i,y} \cdot \hat{\mathbf{y}}_\perp = v_{i,y}|\mathbf{y}_\perp|$.

As explained in Section 8 of Richmond (1995), the mapping of measured convection velocities (or equivalently electric fields by virtue of Equation 17) from the measurement altitude to the reference altitude $h_R = 110$ km is handled by the definitions of the MA-110 basis vectors $\mathbf{d}_1$, $\mathbf{d}_2$, $\mathbf{e}_1$, and $\mathbf{e}_2$. In summary our representation of the ionospheric potential $\Phi$ defined in Section 3.1 is constant along magnetic field lines, as are its partial derivatives in Equation 16, along with the coefficients $E_{d1}$
and $E_{d2}$ in Equation 16 and $v_{e1}$ and $v_{e2}$ in Equation 18. Consequently all dependence on altitude is contained in the definition of the MA-110 basis vectors, and information about the mapping from the Swarm altitude of measurement to $h_R$ is represented by the dot products $\hat{\mathbf{y}} \cdot \mathbf{e}_1$ and $\hat{\mathbf{y}} \cdot \mathbf{e}_2$ in Equation 19.

### 3.1    Definition of potential $\Phi$ and model coefficients

Assuming the ionospheric electric potential $\Phi$ does not vary along magnetic field lines (i.e., field lines are equipotentials) we
may represent $\Phi$ via Equation A3 in Laundal et al. (2018):

$$\Phi\left(\lambda, \phi\right) = R_E \sum_{n,m} P_n^m\left(\mu\right)\left[g_n^m \cos\left(m\phi\right) + h_n^m \sin\left(m\phi\right)\right], \tag{21}$$

where $P_n^m$ are the Schmidt semi-normalized associated Legendre functions, and $\mu = \sin\lambda$. We use MKS units such that the coefficients $g_n^m$ and $h_n^m$ are given in V/m. Expressing $\Phi$ in this way assumes that the (non-orthogonal) MA-110 coordinate system is spherical. While we cannot rigorously justify this assumption, we point the interested reader to the numerical test
performed by Laundal et al. (2016) in which they calculate the total energy content of spherical harmonic fits to a synthetic



magnetic potential represented in both geocentric spherical coordinates and (assumed orthogonal) Apex coordinates, and find that the total energy content differs by less than 0.1% between the two representations.

We wish to derive the Swarm Hi-C model in as consistent a manner as possible with the derivation of the AMPS model. We follow Laundal et al. (2018) in expanding each spherical harmonic coefficient, for example $g_n^m$, as a function of 19 external parameters:

$$
\begin{aligned}
g_n^m = & \, g_{n0}^m + g_{n1}^m \sin\theta_c + g_{n2}^m \cos\theta_c + g_{n3}^m \epsilon + g_{n4}^m \epsilon \sin\theta_c + g_{n5}^m \epsilon \cos\theta_c + \\
& g_{n6}^m \Psi + g_{n7}^m \Psi \sin\theta_c + g_{n8}^m \Psi \cos\theta_c + g_{n9}^m \Psi\epsilon + g_{n10}^m \Psi\epsilon \sin\theta_c + g_{n11}^m \Psi\epsilon \cos\theta_c + \\
& g_{n12}^m \tau + g_{n13}^m \tau \sin\theta_c + g_{n14}^m \tau \cos\theta_c + g_{n15}^m \Psi\tau + g_{n16}^m \Psi\tau \sin\theta_c + g_{n17}^m \Psi\tau \cos\theta_c + \\
& g_{n18}^m \mathrm{F}_{10.7}.
\end{aligned}
\tag{22}
$$

Here $\Psi$ is the dipole tilt angle, $\theta_c = \arctan 2\,(B_y, B_z)$ is the IMF clock angle,

$$
\epsilon = 10^{-3} |v_x|^{4/3} \left(B_y^2 + B_z^2\right)^{2/3} \sin^{8/3}(\theta_c/2)
\tag{23}
$$

is the Newell et al. (2007) coupling function, with the $x$ component of the GSM solar wind speed in km/s and IMF components (in GSM coords) in nT. The quantity

$$
\tau = 10^{-3} |v_x|^{4/3} \left(B_y^2 + B_z^2\right)^{2/3} \cos^{8/3}(\theta_c/2)
\tag{24}
$$

is defined by analogy with Equation 23, and maximizes for strictly northward IMF. In contrast to, for example, the ionospheric potential models of Weimer (2005) and Zhu et al. (2021), we have not implemented any saturation of the effect of $\epsilon$ or $\tau$ for extreme events.

## 3.2 Constraining the potential $\Phi$ at $\pm 47°$ MA-110 latitude

Swarm TII ion drift measurements are calibrated by setting the average drift between Quasi-Dipole latitudes of $44°$ and $50°$ in the Northern Hemisphere ($-44°$ and $-50°$ in the Southern Hemisphere) to zero (Burchill and Knudsen, 2022). Calibrated measurements of $v_{i,y}$ over these latitudes therefore generally have magnitudes of a few tens of m/s, and we find that these calibrated measurements are effective in constraining the east-west ($v_{e1}$) component of model convection velocities at these latitudes. The model convection in the north-south direction is, in contrast, essentially unconstrained by measurements. We therefore analytically impose the constraint $\Phi(\lambda = \pm 47°, \phi) = 0$ on the model. This constraint forces the approximately east-west electric field component $E_{d1}$ (and therefore the approximately north-south convection velocity component $v_{e2}$) at $\pm 47°$ MA latitude to be zero, and modifies the spherical harmonic expansion given by Equation 21.

Our procedure for imposing this constraint may be summarized as follows. We begin by writing the contribution to the total potential $\Phi$ from a particular spherical harmonic order $m$:

$$
\begin{aligned}
\Phi^m / R_E &= \cos m\phi \sum_{n=n'}^{N} P_n^m(\mu) g_n^m + \sin m\phi \sum_{n=n'}^{N} P_n^m(\mu) h_n^m, \\
&= \cos m\phi \left(\mathbf{P}^m\right)^T \mathbf{g}^m + \sin m\phi \left(\mathbf{P}^m\right)^T \mathbf{h}^m,
\end{aligned}
\tag{25}
$$





where $n' = \max(1, m)$, $\mathbf{P}^m = \left(P_{n'}^m, P_{n'+1}^m, \ldots, P_N^m\right)^T$, and $\mathbf{g}^m = \left(g_{n'}^m, g_{n'+1}^m, \ldots, g_N^m\right)^T$.

Using Equation (A6) from the Appendix, the second line of Equation 25 may be written

$$\Phi^m/R_E = \cos m\phi \, (\mathbf{P}^m)^T \left(\mathbf{A}^m \mathbf{g}^{m'}\right) + \sin m\phi \, (\mathbf{P}^m)^T \left(\mathbf{A}^m \mathbf{h}^{m'}\right), \tag{26}$$

where the matrix $\mathbf{A}^m$ enforces the constraints $\Phi(\lambda = \pm 47°, \phi) = 0$ by specifying the two lowest-degree model coefficients for a particular model order $m$ in terms of the remaining order-$m$ model coefficients of higher degree, represented by $\mathbf{g}^{m'}$ and $\mathbf{h}^{m'}$. The potential (21) can then be expressed as

$$\Phi = \sum_{m=0}^M \Phi^m = R_E \sum_{m=0}^M (\mathbf{P}^m)^T \mathbf{A}^m \left(\mathbf{g}^{m'} \cos m\phi + \mathbf{h}^{m'} \sin m\phi\right). \tag{27}$$

To be consistent with the derivation of the AMPS model, we also use a maximum spherical harmonic degree $N = 65$ and

order $M = 3$, corresponding to a total of 8,531 model coefficients, or $8{,}531/19 = 449$ spherical harmonic coefficients. These constraints reduce the number of independent spherical harmonic coefficients by 14 from 449 to 435: a total of eight $g_n^m$ coefficients (two for each model order $m = 0$–3), and six $h_n^m$ coefficients (two for each model order $m = 1$–3).

### 3.3 Matrix equation

Equation 27 can be related to Equation 19 using the partial derivatives of $\Phi$:

$$\frac{\partial \Phi}{\partial \phi} = R_E \sum_{m=0}^M m (\mathbf{P}^m)^T \mathbf{A}^m \left(-\mathbf{g}^{m'} \sin m\phi + \mathbf{h}^{m'} \cos m\phi\right);$$

$$\frac{\partial \Phi}{\partial \lambda} = R_E \sum_{m=0}^M \frac{\partial (\mathbf{P}^m)^T}{\partial \lambda} \mathbf{A}^m \left(\mathbf{g}^{m'} \cos m\phi + \mathbf{h}^{m'} \sin m\phi\right); \tag{28}$$

where $\frac{\partial P_n^m(\mu)}{\partial \lambda} = \cos \lambda \frac{\partial P_n^m(\mu)}{\partial \mu}$.

Inserting Equations 16 into Equation 19 with partial derivatives of $\Phi$ in the former given by Equations 28, finally yields

$$\hat{\mathbf{y}} \cdot \mathbf{v}_E = \sum_{m=0}^M (\mathbf{x}^m)^T \mathbf{k}^m$$

$$= \mathbf{x}^T \mathbf{k}, \tag{29}$$

with $\mathbf{k}^m = \begin{pmatrix} \mathbf{g}^{m'} \\ \mathbf{h}^{m'} \end{pmatrix}$, $\mathbf{x}^m = \mathbf{x}_1^m + \mathbf{x}_2^m$, and

$$(\mathbf{x}_2^m(\lambda, \phi))^T = \frac{R_E}{(R_E + h_R) B_{e3}} \frac{\hat{\mathbf{y}} \cdot \mathbf{e}_2}{\cos \lambda} m (\mathbf{P}^m)^T \left(-\mathbf{A}^m \sin m\phi \quad \mathbf{A}^m \cos m\phi\right); \tag{30}$$

$$(\mathbf{x}_1^m(\lambda, \phi))^T = \frac{R_E}{(R_E + h_R) B_{e3}} \frac{\hat{\mathbf{y}} \cdot \mathbf{e}_1}{\sin I_m} \frac{\partial (\mathbf{P}^m)^T}{\partial \lambda} \left(\mathbf{A}^m \cos m\phi \quad \mathbf{A}^m \sin m\phi\right). \tag{31}$$

Equation 29 is linear in the model coefficients $g_n^m$ and $h_n^m$, which means that given a sufficient number of measurements of $\hat{\mathbf{y}} \cdot \mathbf{v}_E = \mathbf{v}_{i,y} \cdot \hat{\mathbf{y}}_\perp$ we may solve a matrix equation of the form

$$\mathbf{d} = \mathbf{X} \mathbf{k} \tag{32}$$



for the model coefficients, with $\mathbf{d}$ and $\mathbf{Xk}$ respectively the LHS and RHS of Equation 29 stacked vertically for multiple measurements.

## 3.4 Cost function and inversion procedure

Lowes (1966) (but see also Section 5 in Sabaka et al., 2010) shows that for a potential of the form

$$V = a \sum_{n=1}^{\infty} \left( \frac{R_E}{r} \right)^{n+1} \sum_{m=0}^{n} P_n^m (\cos\theta) [g_n^m \cos m\phi + h_n^m \sin m\phi] \tag{33}$$

the average energy contained in the field $\mathbf{E}_\perp = -\nabla V$ at $r = R_E$ is

$$\langle |\mathbf{E}_\perp|^2 \rangle (\mathbf{k}) = \sum_{n=1}^{\infty} (n+1) \sum_{m=0}^{n} \left[ (g_n^m)^2 + (h_n^m)^2 \right]. \tag{34}$$

We use this result to define a cost function that minimizes both the model-measurement error and total energy of the electric field:

$$\phi = \text{error}(\mathbf{k}) + \kappa \langle |\mathbf{E}_\perp|^2 \rangle (\mathbf{k}), \tag{35}$$

which we then minimize with respect to $\mathbf{k}$ to obtain an initial estimate of the model coefficients:

$$\mathbf{k}_0 = \left( \mathbf{X}^T \mathbf{X} + \mathbf{L} \right)^{-1} \mathbf{X}^T \mathbf{d}. \tag{36}$$

The model regularization matrix $\mathbf{L}$ is zero everywhere except for the diagonal elements $\kappa(n+1)$, which correspond to $(g_n^m)^2$ and $(h_n^m)^2$. This matrix represents a constraint on the total energy contained in the model field, such that (i) the model field energy is the minimum required to fit the Swarm electric field measurements, and (ii) the higher the order $n$ of the model

coefficient, the more regularization is applied. The value of $\kappa$ is chosen to be the smallest value for which $\mathbf{X}^T \mathbf{X} + \mathbf{L}$ is invertible, starting with $\kappa = 10^2$ followed by increments of powers of $\sqrt{10}$. Equation (36) is identical in form to Laundal et al. (2018) Equation (A8), although the elements of $\mathbf{L}$ here differ from those given by Laundal et al. (2018).

Regarding the need for regularization, given that the number of measurements (tens of millions) far exceeds the number of model coefficients (thousands), Equation 32 is highly overdetermined. We nevertheless find that the inverse problem is

ill-conditioned in practice, in that convergence is not achieved without some regularization being applied.

After obtaining an initial estimate of the model coefficients, we iteratively reweight the measurements using Huber weights. Quoting from Laundal et al. (2018), "In each step, each equation is reweighted according to how well the corresponding data point fits the model prediction from the previous iteration. The purpose of this procedure [...] is to reduce the effect of outliers, and to enable the final solution to better represent typical values rather than simple mean values."

As previously mentioned we truncate the spherical harmonics expansion at maximum degree $N = 65$ and maximum order $M = 3$. The truncation of $N$ affects the resolution of the model by constraining the wavelength of the spherical harmonic surface waves. That $M$ is much less than $N$ indicates that the sectoral resolution of the model is much lower than than the zonal resolution. This choice reflects both a desire to avoid overfitting and the common understanding that large-scale high-latitude ionospheric currents tend to align along magnetic east/west. Similar choices were made by e.g. Laundal et al. (2018)





and Friis-Christensen et al. (1985) and has an effect similar to regularization of east/west gradients (Madelaire et al., 2023). After ∼20 iterations, the model vector magnitude $|\mathbf{k}|$ changes by less than 3.9% relative to the magnitude of the initial estimate $|\mathbf{k}_0|$. Given that Laundal et al. (2018) used a threshold of 1% to terminate their iterative estimation of AMPS model coefficients this percentage may seem high, but the design of our model is such that the lowest-degree model coefficients, which typically have the largest magnitudes by virtue of the regularization of $\langle E^2 \rangle$ in Equation 34, are precisely the coefficients that are

eliminated by the constraint matrices $\mathbf{A}^m$ in Equations 29 and 32. These coefficients are not included in the calculation of $|\mathbf{k}|$ during each iteration, because they depend on the values of all other coefficients (cf. Equations A3 and A5) and on the model input parameters listed in Equation 22. In practice, the model predictions for model coefficients produced after reaching a relative change of ∼4% (i.e., after iteration 20) are very similar; for example, the cross-polar cap potential values discussed in Section 4 and shown in each panel of Figures 2–4 typically vary by less than 0.01 kV.

We also derived a model with maximum degree and order of respectively $N = 60$ and $M = 5$, which had 11,742 model coefficients as opposed to 8,531 model coefficients for the $N = 65$, $M = 3$ model. The overall reduction in average misfit with this expanded model was less than 0.1%.

## 4   Swarm Hi-C model results

Here we compare Northern and Southern Hemisphere potentials for different IMF clock angles and dipole tilts, and compare
Swarm Hi-C cross-polar cap potential (CPCP) values with CPCP values reported previously.

### 4.1   Comparison of ionospheric potentials in each hemisphere

Figure 2 shows the ionospheric potentials in the NH (colored contours) over 50–90° MLat for a transverse IMF component $B_T = \sqrt{B_y^2 + B_z^2} = 5$ nT and a solar wind speed $v_{SW} = 450$ km/s during local winter ($\Psi = -25°$). To facilitate assessment of how well the assumption of mirror symmetry (Equation 1) holds, the ionospheric potentials in the SH are also shown (black
contour lines) with the signs of $B_y$ and $\Psi$ reversed. We have chosen these solar wind and IMF conditions to approximately match those chosen in the other studies whose reported cross-polar cap potential (CPCP) values we compare with in Section 4.2 (see Table 1).

     The assumption of mirror symmetry generally holds well for potential patterns in the two hemispheres during local winter, but there are also relatively large deviations. These are most apparent in the panels for which the ratio of CPCP values in
each hemisphere depart from 1 by more than a few percent—for example, for $\theta_c = 0°$ ($B_z = 5$ nT, $B_y = 0$) and $\theta_c = 135°$ ($-B_z = B_y = 5/\sqrt{2}$ nT). For the eight panels shown, the average CPCP value for both the NH and SH is 51 kV.

     Figures 3 and 4 show the ionospheric potentials in the two hemispheres during equinox ($\Psi = 0°$) and local summer ($\Psi = 25°$ for the NH and $\Psi = -25°$ for the SH), for the same solar wind and IMF conditions used in Figure 2. As during local winter, during equinox and local summer the most apparent differences in the convection patterns between hemispheres occur for $\theta_c$
orientations having a positive $B_z$ component. The four-cell lobe reconnection pattern is also visible in both hemispheres for $\theta_c = 0°$ during local summer (top center panel in Figure 4).





**Figure 2.** Ionospheric potential in the Northern Hemisphere (colored contours) and Southern Hemisphere (black contour lines) as a function of IMF clock angle for dipole tilt angle $\Psi = \mp 25°$ (local winter), where the signs of $B_y$ and $\Psi$ are reversed for the Southern Hemisphere (local winter). The spacing between contours and contour lines is constant to facilitate comparison between hemispheres, and is such that no more than seven contour levels are shown in each panel. The spacing therefore varies from panel to panel. The colors of all contour levels are nonetheless scaled to the colorbar shown at bottom. In this figure, the mean of all eight CPCP values (shown in top right corner of each panel) for the Southern Hemisphere is 3% greater than for the Northern Hemisphere.







**Figure 3.** Ionospheric potential in the Northern Hemisphere (colored contours) and Southern Hemisphere (black contour lines) as a function of IMF clock angle for dipole tilt angle $\Psi = 0°$ ($\sim$equinoxes), in the same layout as Figure 2. In this figure, the mean CPCP (top right corner of each panel) for the Northern Hemisphere is 5% greater than for the Southern Hemisphere.





**Figure 4.** Ionospheric potential in the Northern Hemisphere (colored contours) and Southern Hemisphere (black contour lines) as a function of IMF clock angle for dipole tilt angle $\Psi = \pm 25°$ (local summer), in the same layout as Figures 2 and 3. In this figure, the mean CPCP (top right corner of each panel) for the Northern Hemisphere is 13% greater than for the Southern Hemisphere.





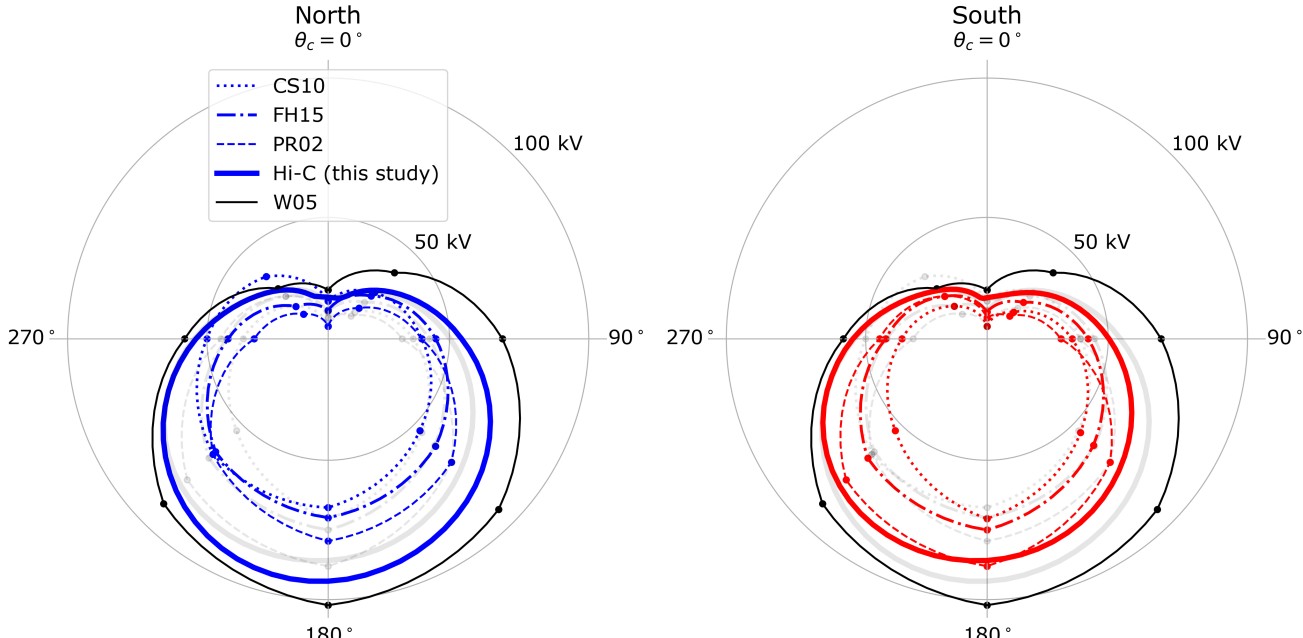

**Figure 5.** Cross-polar cap potential in the Northern (left) and Southern (right) Hemisphere as a function of IMF clock angle $\theta_c$. The clock angle is defined such that $\theta_c = \{0°, 90°, 180°\}$ correspond respectively to $\{B_z > 0, B_y > 0, B_z < 0\}$ nT, and so forth. Results from the Swarm Hi-C model (this study) are shown as thick solid lines. The other studies included for comparison are shown in the caption. Table 1 gives the solar wind and dipole tilt conditions used in each study. To facilitate comparison between hemispheres, the light gray line with matching line style in each panel shows CPCP values from the opposite hemisphere for the same model; the only exception is CPCP values from the W05 model (thin black line), which does not distinguish between hemispheres. CS10: Cousins and Shepherd (2010); FH15: Förster and Haaland (2015); PR02: Papitashvili and Rich (2002); W05: Weimer (2005).

### 4.2 Cross-polar cap potential comparison

Figure 5 compares cross-polar cap potential values derived from the Hi-C model (thick solid lines) with results from the four other studies listed in Table 1, which also gives the solar wind and dipole tilt conditions used for deriving CPCP values in each

study. In the four previous studies referenced in this figure, CPCP estimates were only given for the eight IMF clock angles indicated by the dots. Hence the curves for these studies are obtained by linear interpolation between these dots, with the linear interpolation carried out in polar coordinates. This is the cause of the cusp-like feature at $\theta_c = 0°$ for the curves representing these studies. This cusp-like feature does not appear in the Swarm Hi-C curve (thick solid line) in each panel because these curves are directly calculated from the Hi-C model at a resolution of $\Delta\theta_c = 0.1°$.

In general Swarm Hi-C CPCP values are similar to those yielded by other models. The SuperDARN-based model of Cousins and Shepherd (2010) generally yields the smallest CPCP, but Thomas and Shepherd (2018) have shown that convection mea-



|  | $v_{sw}$ | $B_T$ | $n_{sw}$ | $\Psi$ | Comments |
|---|---|---|---|---|---|
|  | [km/s] | [nT] | [cm$^{-3}$] | [deg] |  |
| Papitashvili and Rich (2002) | 400 | 5 | 5 | $\sim 0$ | Meas. in 2-mo. window around equinox |
| Weimer (2005) | 450 | 5 | 4 | 0 |  |
| Cousins and Shepherd (2010) | N/A | N/A | N/A | Note$^*$ | $2.2 < E_{sw} < 2.9$ mV/m$^{**}$ |
| Förster and Haaland (2015) | N/A | N/A | N/A | N/A | All data w/ bias vector length $> 0.96^{***}$ |
| This study | 450 | 5 | N/A | 0 |  |

$^*\Psi > 10°$ NH, $\Psi < -10°$ in SH.

$^{**}E_{sw} = |V_x B_T| = 2.5$ mV/m for $V_x = -500$ km/s and $B_T = 5$ nT, for example.

$^{***}$cf. their Section 3.

**Table 1.** Solar wind, IMF, and dipole tilt conditions used by studies shown in Figure 5 to produce CPCP estimates.

surements from the mid-latitude SuperDARN radars, which have become available more recently and were not used by Cousins and Shepherd (2010), are important to get the correct potential during intense solar wind driving.

## 5  Swarm Ionospheric Polar Electrodynamics (Swipe): the combined AMPS and Swarm Hi-C models

In this section we present a comparison of Swipe estimates of electromagnetic work and Hall and Pedersen conductances in each hemisphere. This comparison is carried out using the same dipole tilt, solar wind, and IMF conditions as were used in Section 4.1. The same figure format is also used, with NH distributions plotted as contours and SH distributions as black contour lines, and the sign of IMF $B_y$ and dipole tilt $\Psi$ inverted for SH distributions to facilitate assessment of mirror symmetry (see Equation 1). Distributions of Swipe estimates in this section are however only shown over 60–90° MLat in the NH ($-60$

to $-90°$ MLat in the SH) since equatorward of $\pm 60°$ MLat the estimates of electromagnetic work are essentially zero, and the estimates of Hall and Pedersen conductances are typically invalid as described below.

   Some preliminary comments are in order. In Section 2 we arrived at the estimates of Hall and Pedersen conductances given by Equations 12–13 by assuming that the electric field in the reference frame of the neutral wind $\mathbf{E}'_\perp = \mathbf{E} + \mathbf{v}_n \times \mathbf{B}$ does not substantially differ from the electric field $\mathbf{E}$ in an Earth-fixed frame of reference. Under this assumption Equation 5 indicates

that $w = w_J \geq 0$, and correspondingly that $W = \int w \, dr = \int w_J \, dr \geq 0$. We may therefore be confident that at any location where either the height-integrated electromagnetic work $W < 0$ or the Hall conductance $\Sigma_H < 0$, Swipe estimates of $\Sigma_H$ and $\Sigma_P$ are either inconsistent with this assumption, or are related to differences between the Swarm Hi-C and AMPS models despite our best attempt to derive them in a consistent fashion. In Figures 10–15 we therefore only show portions of the distributions of $\Sigma_H$ and $\Sigma_P$ where the following criteria are met:





$\mathbf{J}_\perp \cdot \mathbf{E}_\perp \geq 0.5 \ \mathrm{mW/m}^2;$

$\Sigma_H > 0.$                (37)

The first criterion is most important in that where it is not met (typically within the polar cap and equatorward of $\pm 60°$ MLat) the conductance estimates are in many places negative or unphysically large, or exhibit sharp gradients. The second criterion is primarily relevant above $70°$ MLat where the Hall conductance estimates are in some places negative (typically no less than $-1$ mho but for some tilt/solar wind/IMF configurations as low as $-6$ mho). The threshold 0.5 mW/m$^2$ is obtained via a

rough estimate of the typical contribution of the height-integrated second term on the RHS of Equation 5, $\mathbf{v}_n \cdot (\mathbf{J}_\perp \times \mathbf{B})$, given typical values $|\mathbf{v}_n| = 100$ m/s, $|\mathbf{B}| = 5 \times 10^5$ nT, and $|\mathbf{J}_\perp| = 100$ mA/m. In practice we find that for threshold values below 0.5 mW/m$^2$ very sharp gradients in the conductance distributions appear.

We also note that model predictions represent "average" large-scale electrodynamics for a given set of model input parameters, and that ideally one would also take stock of the uncertainty of model predictions in assessing physical consistency. None

of the studies referenced in Table 1 address model uncertainty, nor do we directly address it in this study. This topic deserves more attention as part of a dedicated study.

### 5.1   $\mathbf{J}_\perp \cdot \mathbf{E}_\perp$ work

Figures 6–8 show distributions of electromagnetic work given by Equation 11. As with the electric potentials in Figures 2–4, the distributions of electromagnetic work in the two hemispheres are in general highly similar. The largest differences appear

during local summer for $\theta_c$ orientations with a negative IMF $B_z$ component (bottom row of panels in Figure 8), for which the NH distributions are overall more intense and electromagnetic work in the polar cap is greater, consistent with the CPCP values shown in the bottom row of Figure 4. On the other hand the hemispheric ratios of integrated work (indicated in the top right corner of each panel) are largest for $\theta_c$ orientations having a positive IMF $B_z$ component, although the magnitudes of integrated work are generally very small for these $\theta_c$ orientations.

In each panel of each of these figures the electromagnetic work integrated over the entire polar ionosphere for each hemisphere is indicated in the top right corner. Comparison of these values for a given IMF orientation and different seasons indicates that the integrated electromagnetic work strictly increases from local winter to local summer (where equinox is strictly between winter and summer): During local winter conditions ($\Psi = \mp 25°$, Figure 6) integrated work ranges between 1 and 46 GW, whereas during local summer conditions ($\Psi = \pm 25°$, Figure 8) integrated work ranges between 7 and 108 GW.

For all seasons and for IMF $B_z > 0$ configurations (top three panels in each figure) the locus of enhanced work on the dayside depends on the sign of IMF $B_y$. In particular, in the top left corner ($B_y < 0$ in the NH, $B_y > 0$ in the SH) the enhancement is greatest over post-noon MLTs, and in the top right corner ($B_y > 0$ in the NH, $B_y < 0$ in the SH) the enhancement is greatest over pre-noon MLTs on the dayside. This dependence on IMF $B_y$ is the same as the dependence exhibited by Alfvénic energy deposition (e.g., Figure 2 in Hatch et al., 2018), and opposite the dependence exhibited by the polar cusp (Zhang et al., 2013, and references therein). Regarding the latter, for increasingly negative $B_y$ the NH polar cusp tends to shift to increasingly early






**Figure 6.** Electromagnetic work in the Northern Hemisphere (colored contours) and Southern Hemisphere (black contour lines) as a function of IMF clock angle for dipole tilt angle $\Psi = \mp 25°$ (local winter), in the same layout as Figure 2. The spacing between contours and contour lines is constant to facilitate comparison between hemispheres, and is such that no more than four contour levels are shown in each panel. The spacing therefore varies from panel to panel. The color of all contour levels are nonetheless scaled to the colorbar shown at bottom.





**Figure 7.** Electromagnetic work in the Northern Hemisphere (colored contours) and Southern Hemisphere (black contour lines) as a function of IMF clock angle for dipole tilt angle $\Psi = 0°$ (equinox), in the same layout as Figure 2.





**Figure 8.** Electromagnetic work in the Northern Hemisphere (colored contours) and Southern Hemisphere (black contour lines) as a function of IMF clock angle for dipole tilt angle $\Psi = \pm 25^{\circ}$, in the same layout as Figure 2.





MLTs, and vice versa for increasingly positive $B_y$ (Zhou et al., 2000; Zhang et al., 2013). (Note however the report of Wing et al., 2001, on the existence of a double cusp during strong, dominant $B_y$ solar wind conditions.)

In contrast, the maps of electric potential in Figures 2–4 indicate no clear pre- or post-noon asymmetry of the magnitude of dayside ionospheric flows inside the polar cap, depending on the sign of IMF By, as mentioned in the previous paragraph discussing the maps of electromagnetic work. From looking at the corresponding maps of the horizontal ionospheric currents during the same conditions in Figures 9–11 in Laundal et al. (2018), it is evident that the asymmetry in the electromagnetic work seen is related to the asymmetries in the horizontal currents. The mentioned opposite asymmetry compared to the cusp location can be understood by the direction of the direct forcing from the IMF due to a dominant $B_y$ component, which tends to have a more direct influence during local summer conditions (Reistad et al., 2021b). We note that for a neutral wind field corotating with the Earth, this direct $B_y$ forcing on the dayside will for IMF $B_y > 0$ in the NH go against the corotation wind field, while for IMF $B_y < 0$ in the NH it will point along the corotation, reducing the electric field in the neutral wind frame. This effect may be an important cause of the asymmetries pointed out here in the electromagnetic work and horizontal current maps in the dayside polar cap, especially during local summer.

Regardless of IMF $B_y$ and dipole tilt $\Psi$, integrated electromagnetic work tends to increase with increasingly negative IMF $B_z$ as is well known from previous studies (e.g., Figure 5 in Weimer, 2005). The ratio of integrated electromagnetic work in the NH and SH, respectively $W_N$ and $W_S$, for $\theta_c$ orientations involving a negative IMF $B_z$ component (bottom three panels in Figures 6–8) shows a general tendency to increase from local winter ($W_N/W_S = 1.03$–$1.08$) to local summer ($W_N/W_S = 1.15$–$1.25$). There is also a general tendency for integrated work to be greater for IMF $B_y > 0$ in the NH (IMF $B_y < 0$ in the SH), with some exceptions visible for the NH in the bottom row of Figure 8 and in the top rows of Figures 6–7.

All of the foregoing figures are based on IMF $B_T = 5$ nT. To elucidate the relationship between the magnitude of IMF $B_T$ and integrated NH and SH electromagnetic work (respectively $W_N$ and $W_S$) for different seasons, the top three rows of Figure 9 shows $W_N$ (left column) and $W_S$ (middle column) as functions of $B_T$ and IMF clock angle $\theta_c$, with the sign of $B_y$ and dipole tilt $\Psi$ reversed according to Equation 1. We also define a hemispheric asymmetry coefficient $A_{EM} = 2(W_N - W_S)/(W_N + W_S)$, which is shown in the right column. The bottom row of Figure 9 shows $A_{EM}$ averaged over $\theta_c$ as a function of $B_T$ (left) and averaged over $B_T$ as a function of $\theta_c$.

The first two columns of the top three rows show that $W_N$ and $W_S$ tend to maximize for $\theta_c$ configurations dominated by negative IMF $B_z$, as expected, and there is an overall trend toward increasing $W_N$ and $W_S$ for increasing $B_T$. The right column of the top three rows shows that during equinox and local summer conditions the asymmetry coefficient $A_{EM}$ also tends to maximize for $\theta_c$ configurations dominated by negative IMF $B_z$. In contrast, during local winter $A_{EM}$ instead maximizes for $\theta_c$ configurations dominated by positive IMF $B_z$. It is however apparent from Figure 6 that during local winter for such $\theta_c$ configurations $W_N$ and $W_S$ are typically no more than a few GW, which is approximately the same order of magnitude as the uncertainty of $W_N$ and $W_S$. The reliability of $A_{EM}$ during local winter for positive $B_z$-dominated configurations is therefore unclear, although we note the finding of (Workayehu et al., 2021) that the largest NH/SH asymmetries in field-aligned and ionospheric currents occur during positive $B_z$ and $B_y$ during local winter and fall. Regardless, for $\theta_c$ between approximately $90°$ and $270°$ $A_{EM}$ is mostly between 0.05 and 0.3, which corresponds to $W_N/W_S = 1.05$–$1.35$. This confirms that the trend





**Figure 9.** Top three rows: Integrated hemispheric electromagnetic work in the Northern Hemisphere (left column, $W_N$) and Southern Hemisphere (middle column, $W_S$), as well as hemispheric asymmetry coefficient (right column), as a function of transverse IMF component magnitude $B_T$ ($y$ axis) and IMF clock angle $\theta_c$ ($x$ axis). Results for the Southern Hemisphere are shown with the sign of IMF $B_y$ and dipole tilt $\Psi$ reversed. Results for local winter, equinox, and local summer conditions are respectively shown in the top, middle, and bottom rows. The colorbar for $W_N$ and $W_S$ is in GW, and the colorbar for $A_{EM}$ is unitless. Bottom row: The line plots at bottom show the hemispheric asymmetry coefficient $A_{EM}$ from the right column of the top three rows averaged over $\theta_c$ as a function of $B_T$ (left) and averaged over $B_T$ as a function of $\theta_c$; local winter, equinox, and local summer are respectively indicated by the lines labeled "W", "E", and "S".





toward higher values of $W_N/W_S$ moving from local winter to local summer seen in Figures 6–8 applies to a wide range of $B_T$ values.

In summary Figure 9 indicates that (i) $A_{EM}$ is mostly independent of $B_T$; (ii) For negative $B_z$-dominated $\theta_c$ orientations $W_N$ and $W_S$ maximize and $A_{EM} > 0$, with a general tendency for $A_{EM}$ to increase from local winter to local summer for

$B_T >= 1.5$ nT; (iii) Averaging over $\theta_c$, $A_{EM} > 0$ (i.e., the NH is dominant) for most seasons and values of $B_T$; (iv) $A_{EM}$ shows a weak tendency to increase with increasing $B_T$. Regarding the third point, Workayehu et al. (2020) have reported that the strength of NH ionospheric and field-aligned currents tend to be greater than those in the SH almost irrespective of season.

## 5.2 Hall conductance

Figures 10–12 show distributions of Hall conductance in each hemisphere. Regions where the criteria (37) are not met in the

NH are indicated in gray. (Regions where these criteria are not met in the SH are similar to those in the NH, and are shown in Figures S1–S3 of the Supporting Information.) In examining these figures one must observe that within the gray regions where the criteria (37) are not met, the conductances are not necessarily low and indeed may maximize. Furthermore, since the regions where the criteria are met in general reflect the regions where the electromagnetic work exceeds 0.5 mW/m$^2$, the outermost contours in both NH and SH distributions therefore primarily indicate the boundary of where the criteria are met.

These contours are therefore not useful for assessing hemispheric differences.

Bearing the foregoing in mind, we observe a general tendency in predicted distributions of $\Sigma_H$ in both hemispheres to increase with increasingly negative $B_z$-dominated $\theta_c$ configurations, regardless of season, i.e., in all three of Figures 10–12. There is also a general tendency toward increased $\Sigma_H$ on the dayside as season shifts from local winter to local summer.

We now turn to the response of the NH and SH distributions of $\Sigma_H$ on the nightside (18–06 MLT) over auroral latitudes (60–

75° MLat) to different IMF orientations. In the NH, for $\theta_c$ orientations involving either zero or positive $B_y$ and either zero or negative IMF $B_y$ (middle right, bottom right, and bottom center panels in Figures 10–12), both the average Hall conductance and the spatial variability of Hall conductances within this region is highest during winter and lowest during summer. It is primarily during local winter that Hall conductances above 10 mho occur on the nightside in the NH. In contrast, for the $\theta_c$ orientations shown in the middle left and lower left panels of Figures 10–12 (negative $B_y$ in the NH and positive $B_y$ in the

SH, and either zero or negative $B_z$), the average nightside Hall conductance between 60 and 75° MLat is lowest during local winter and highest during local summer in both hemispheres.

In the SH, for all $\theta_c$ orientations for which IMF $B_z$ is zero or negative the average nightside Hall conductance between $-60°$ and $-75°$ MLat is generally less responsive to changes in season, but tends to increase moving from local winter to local summer. The standard deviation of Hall conductances for these $\theta_c$ orientations is likewise lowest during local winter and

highest during local summer.





**Figure 10.** Hall conductance in the Northern Hemisphere (colored contours) and Southern Hemisphere (black contour lines) as a function of IMF clock angle for dipole tilt angle $\Psi = \mp 25°$ (local winter), in the same layout as Figure 2. Areas where the criteria (37) are not met in the Northern Hemisphere are indicated in gray.





**Figure 11.** Hall conductance in the Northern Hemisphere (colored contours) and Southern Hemisphere (black contour lines) as a function of IMF clock angle for dipole tilt angle $\Psi = 0°$ (equinox), in the same layout as Figure 2. Areas where the criteria (37) are not met in the Northern Hemisphere are indicated in gray.





**Figure 12.** Hall conductance in the Northern Hemisphere (colored contours) and Southern Hemisphere (black contour lines) as a function of IMF clock angle for dipole tilt angle $\Psi = \pm 25°$ (local summer), in the same layout as Figure 2. Areas where the criteria (37) are not met in the Northern Hemisphere are indicated in gray.





**Figure 13.** Pedersen conductance in the Northern Hemisphere (colored contours) and Southern Hemisphere (black contour lines) as a function of IMF clock angle for dipole tilt angle $\Psi = \mp 25°$ (local winter), in the same layout as Figure 2. Areas where the criteria (37) are not met in the Northern Hemisphere are indicated in gray.





**Figure 14.** Pedersen conductance in the Northern Hemisphere (colored contours) and Southern Hemisphere (black contour lines) as a function of IMF clock angle for dipole tilt angle $\Psi = 0°$ (equinox), in the same layout as Figure 2. Areas where the criteria (37) are not met in the Northern Hemisphere are indicated in gray.







**Figure 15.** Pedersen conductance in the Northern Hemisphere (colored contours) and Southern Hemisphere (black contour lines) as a function of IMF clock angle for dipole tilt angle $\Psi = \pm 25°$ (local summer), in the same layout as Figure 2. Areas where the criteria (37) are not met in the Northern Hemisphere are indicated in gray.



## 5.3 Pedersen conductance

Figures 13–15 show distributions of Pedersen conductance in each hemisphere. As in Figures 10–12, regions where the criteria (37) are not met in the NH are indicated in gray. The same word of caution in examining the distributions of Hall conductances applies to examination of the distributions Pedersen conductances.

For purely positive IMF $B_y$ (middle right) and for all three orientations of $\theta_c$ involving a negative $B_z$ component (three bottom panels), the average nightside Pedersen conductance over auroral latitudes in the NH is highest during local winter (Figure 13) and lowest during local summer (Figure 15). The exception is purely negative IMF $B_y$ (middle left), for which the nightside Pedersen conductance shows a slight tendency to increase moving from local winter to local summer

     In contrast, over the corresponding region in the SH the average Pedersen conductance is lowest during local winter and

highest during local summer; the interested reader is referred to Figures S4–S6 in the Supporting Information, where the variation in the SH distributions of Pedersen conductance is shown more clearly than in Figures 13–15.

     We also observe that on the dayside, the Swipe model predicts that the highest Pedersen conductances tend to occur at post-noon MLTs poleward of $\pm 70°$, particularly for positive IMF $B_z$ and negative IMF $B_y$ in the NH (positive IMF $B_y$ in the SH). Dayside Pedersen conductances also tend to be higher for negative IMF $B_y$ in the NH (positive IMF $B_y$ in the SH) regardless

of the sign of IMF $B_z$. These enhanced Pedersen conductances could be related to the frequently appearing afternoon hot spot in Joule heating reported by (Cai et al., 2016).

## 6   Discussion

The goal of this study is to determine to what extent key descriptors of ionosphere-thermosphere electrodynamics, such as the ionospheric potential, the cross-polar cap potential, electromagnetic work, and ionospheric conductances, obey the mirror

symmetry condition given by Equation 1. To achieve this goal we have developed a new empirical model of ionospheric convection based on Swarm TII measurements in an Earth-fixed frame, and combined the outputs of this model with outputs from the empirical AMPS model that is based on Swarm and CHAllenging Mini-satellite Payload (CHAMP) magnetometer measurements.

     The only other study of which we are aware that presents global empirical models of these descriptors of IT electrodynamics

is the work of Weimer and Edwards (2021) (hereafter WE21). An important difference between the Swarm Hi-C and AMPS models and the empirical models that they use is that the former do not assume any form of hemispheric asymmetry, whereas the ionospheric potential model used by WE21 does assume hemispheric mirror symmetry, and the Weimer (2013) model of divergence-free currents used by WE21 is based solely on NH ground magnetometer measurements. Thus while the results presented by WE21 represent an important step toward a fuller understanding of high-latitude IT electrodynamics, the empirical

models they use cannot be employed for testing the degree to which the abovementioned descriptors of IT electrodynamics exhibit mirror symmetry between hemispheres.

     In Section 4.1 we found that the most apparent deviations from mirror symmetry between the two hemispheres tend to occur under $\theta_c$ orientations for which $B_z > 0$. There is also a general tendency for the CPCP in the NH to exceed the CPCP in the



SH by several percent, as shown primarily in Figure 5 but also in Figures 2–4. Since Apex coordinates take stock of geometric differences such as the different polar cap areas in the two hemispheres, these deviations may be attributable to real differences in polar cap convection speeds due to hemispheric differences in, for example, lobe reconnection, as described by both Reistad et al. (2021a) and Pettigrew et al. (2010) and references in those studies. Regardless of the explanation, the CPCP is primarily useful as a general diagnostic, the interpretation of which can be complicated.

Our finding that the potentials in the two hemispheres do not exactly obey the mirror symmetry condition (1) is not new; it has been pointed out by at least Pettigrew et al. (2010) and Förster and Haaland (2015). One important difference between these two earlier studies and our results, however, is that whereas Pettigrew et al. (2010) and Förster and Haaland (2015) both find that the SH CPCP exceeds the NH CPCP for purely negative IMF $B_z$ conditions—in the former study by 0–12 kV, and in the latter by $\sim$5 kV—we find that for purely negative IMF $B_z$ conditions the SH CPCP only exceeds the NH CPCP during local winter.

Besides the comparison of distributions of electromagnetic work that we have carried out in Section 5, we are not aware of any work that directly examine how well the assumption of mirror asymmetry holds for electromagnetic work in each hemisphere. We have concluded on the basis of Figures 6–8 that mirror symmetry mostly holds in the two hemispheres, with hemispheric differences having more to do with differences in the intensity of the distributions of electromagnetic work rather than differences in the shapes of the distributions.

Several related studies that instead examine hemispheric asymmetries in Poynting flux have been performed (Cosgrove et al., 2022; Pakhotin et al., 2021; Knipp et al., 2021); each presents evidence that the electromagnetic energy input to the high-latitude ionosphere is on average greater in the NH than in the SH. These studies are of relevance to this study because of the connection between the divergence of Poynting flux and electromagnetic work given by Poynting's theorem (e.g. Thayer and Semeter, 2004, and references therein) and the "Equipotential Boundary Poynting Flux theorem" presented by Richmond (2010). Of particular relevance to the present study is the finding of Cosgrove et al. (2022) that the overall preference for electromagnetic energy input into the NH may be reversed during local winter. While the Swipe model does not yield evidence in direct support of the hypothesis, the Swipe model does predict that the hemispheric imbalance of energy input is least during local winter for $\theta_c$ orientations dominated by negative IMF $B_z$ (bottom right panel of Figure 9).

While it would be natural to present Swipe model distributions of Poynting flux and compare them to Swipe model distributions of electromagnetic work, an exercise suggested by Richmond (2010) and carried out using synthetic data and empirical models by Vanhamäki et al. (2012) and WE21 respectively, we plan such a comparison for a possible future study.

Regarding the role of neutral winds, we find that the criteria given in Equation 37 allow us to determine locations where Swipe model predictions are inconsistent with the assumption that $\mathbf{v}_n = 0$ in an Earth-fixed frame. When these criteria are met it does not necessarily mean that Swipe model predictions are consistent with this assumption, but they seem sufficient to identify some trends in NH and SH distributions of Hall and Pedersen conductances predicted by the Swipe model (Figures 10–15) which we summarize below. If we do not enforce these criteria, we find that the conductance distributions predicted by Swipe suffer from the same issues that the small selection of conductance distributions presented by Weimer and Edwards (2021) are subject to, namely negative or unphysically large conductances and sharp conductance gradients.





Perhaps the most basic characteristic of distributions of conductance not attributable to solar EUV radiation that have been
presented in previous studies is that the Hall and Pedersen conductances are highest on the nightside (Ahn et al., 1998; Mc-
Granaghan et al., 2015; Hardy et al., 1987; Wallis and Budzinski, 1981). Results from a subset of these previous studies also
suggest a tendency for Hall conductances to be highest over post-midnight MLTs, while the locations of the highest Pedersen
conductances tend to be shifted to relatively earlier local times (Ahn et al., 1998; Hardy et al., 1987).

These characteristics are also present in the conductance distributions predicted by Swipe, primarily for $\theta_c$ orientations
involving predominant IMF $B_y$ or negative IMF $B_z$ and regardless of season, and in the conductance distributions presented
by WE21 for $B_z = -10$ nT. This is noteworthy, as Swipe and WE21 conductance distributions do not distinguish between
contributions to the conductances from solar EUV and auroral precipitation. (We note, however, that the nightside peaks in
Pedersen conductance distributions presented by WE21 partially coincide with negative Hall conductances.) There is a general
tendency for Swipe model conductances on the dayside to increase from local winter to local summer, as expected based on
the increasing exposure to solar EUV radiation from local winter to local summer.

Regarding the tendency of Swipe-predicted NH nightside Hall conductance to decrease moving from local winter to local
summer, this prediction seems to be in line with the finding that the occurrence of discrete aurora is suppressed by sunlight
(Newell et al., 1996). If this interpretation is correct, it is unclear why Swipe-predicted average SH nightside Hall conductances
do not change with season. This model prediction deserves further investigation, although is likely difficult to test given the
general sparsity of SH measurements, both in terms of sheer numbers and in terms of available measurement types.

Regarding our methodology, in implementing the $\Phi(\lambda = \pm47°, \phi) = 0$ analytic constraint in Section 3.2 we chose to repre-
sent the lowest-degree model coefficients $n'$ and $n' + 1$ for a particular order as a sum of the remaining higher-degree model
coefficients (see Equations A3 and A5). This was done because the constraint (34) placed on the E-field power is such that
power in higher-degree coefficients is more constrained than power in coefficients of lower degree. (For example, the amount
of regularization applied to $N = 65$ terms is $\frac{65+1}{3+1} = 66/4 \approx 16.5$ times more than the regularization applied to $N = 3$ terms.)
The higher-degree coefficients therefore tend to be smaller in magnitude than the lower-degree coefficients, unless the mea-
surements dictate otherwise.

On this basis we hypothesized that if we had instead represented the highest-degree coefficients $n = N - 1$ and $n = N$ for a
particular order as sums of the remaining $N - n' + 1$ lower-degree coefficients using expressions analogous to Equations A3
and A5, the magnitude of the resulting model coefficients for degrees $n = N - 1$ and $n = N$ would have been too large and
introduced high-amplitude meridional oscillations into the potential maps. We derived an according alternative set of model
coefficients and used them to produce potential maps similar to those shown in Figures 2–4. We confirmed that the electric
potential patterns were contaminated by high-amplitude meridional oscillations that corresponded to the spherical harmonics
of degree $n = N - 1$ and $n = N$.





## 7 Conclusions

In this study we have presented a new set of empirical models for describing variations in ionosphere-thermosphere electrodynamics in both hemispheres, as a function of season as well as prevailing solar wind and interplanetary magnetic field conditions. These models are primarily based on measurements of magnetic field perturbations and ionospheric plasma drift made by the Swarm satellites. The chief advantage of these models is that they are the first empirical models of high-latitude ionospheric electrodynamics quantities in both hemispheres that are consistently derived in the sense that they (i) take stock of distortions of the Earth's magnetic field via our use of Apex coordinates; (ii) do not assume any form of hemispheric symmetry; and (iii) are based on sets of measurements with similar data coverage distributions. Both the model forward (Hatch and Laundal, 2023a) and inverse (Hatch and Laundal, 2023b) codes are open source and publicly available.

Using these models, we find that model predictions of high-latitude ionospheric potentials and distributions of electromagnetic work in each hemisphere evince a high degree of symmetry when the signs of IMF $B_y$ and dipole tilt $\Psi$ are reversed. In contrast, model predictions of distributions of ionospheric conductances exhibit IMF- and season-dependent hemispheric asymmetries. Ionospheric conductances are generally highest on the nightside. Predicted distributions of ionospheric conductances exhibit very sharp gradients and/or are negative where the magnitude of the electromagnetic work is small. In these areas the assumption that the electric field field in the reference frame of the neutral wind does not substantially differ from the electric field in in an Earth-fixed frame of reference may break down.

*Code and data availability.* The model forward (Hatch and Laundal, 2023a) and inversion (Hatch and Laundal, 2023b) codes are open source and publicly available. The Level 1B Swarm TII data set are publicly accessible at https://swarm-diss.eo.esa.int/#swarm/Advanced/Plasma_Data/2Hz_TII_Cross-track_Dataset. The Penticton Solar Radio Flux at 10.7 cm (F10.7 index) is available at https://lasp.colorado.edu/lisird/data/penticton_radio_flux. Solar wind and IMF measurements are available via the NASA OMNI database: https://omniweb.gsfc.nasa.gov/form/dx1.html.

## Appendix A: Derivation of analytic constraint matrix $\mathbf{A}^m$

It is straightforward to show via Equation 25 that $\Phi^m(\mu_+) = 0$ can be enforced by setting

$$g_{n'}^m = -\sum_{n=n'+1}^{N} \tilde{P}_n^m g_n^m \tag{A1}$$

with $\tilde{P}_n^m = P_n^m(\mu_+)/P_{n'}^m(\mu_+)$. In other words, we can rewrite the lowest-degree (for a given value of the order $m$) coefficient $g_{n'}^m$ in terms of the remaining $N - n'$ higher-degree coefficients, and similarly for $h_{n'}^m$. Inserting (A1) into (25) and rearranging reduces the number of terms in each series in (25) by one:

$$\Phi^m/R_E = \cos m\phi \sum_{n=n'+1}^{N} Q_n^m(\mu)g_n^m + \sin m\phi \sum_{n=n'+1}^{N} Q_n^m(\mu)h_n^m, \tag{A2}$$



with $Q_n^m(\mu) = P_n^m(\mu) - \tilde{P}_n^m P_{n'}^m(\mu)$. In particular $Q_n^m(\mu_+) = 0$.

To enforce $\Phi^m(\mu_-) = 0$ we proceed from Equation A2 in analogous fashion, obtaining

$$g_{n'+1}^m = -\sum_{n=n'+2}^{N} \tilde{Q}_n^m g_n^m \qquad (A3)$$

and a similar expression for $h_{n'+1}^m$, with $\tilde{Q}_n^m = Q_n^m(\mu_-)/Q_{n'+1}^m(\mu_-)$. Inserting these expressions into (A2) then yields

$$\Phi^m/R_E = \cos m\phi \sum_{n=n'+2}^{N} R_n^m(\mu)g_n^m + \sin m\phi \sum_{n=n'+2}^{N} R_n^m(\mu)h_n^m, \qquad (A4)$$

with $R_n^m(\mu) = Q_n^m(\mu) - \tilde{Q}_n^m Q_{n'+1}^m(\mu)$. Note that $R_n^m(\mu_+) = R_n^m(\mu_-) = 0$.

While we do not use expressions (A2) and (A4) to calculate the potential model coefficients, we present them to illustrate how enforcing $\Phi(\mu_\pm) = 0$ effectively reduces the number of terms in each series in Equation 25 by two. To calculate the model coefficients, we insert the expression for $g_{n'+1}^m$ in Equation A3 into Equation A1 to obtain

$$g_{n'}^m = \sum_{n=n'+2}^{N} S_n^m g_n^m \qquad (A5)$$

with $S_n^m = \tilde{P}_{n'+1}^m \tilde{Q}_n^m - \tilde{P}_n^m$.

Using Equations A3 and A5 we may rewrite the vector of order-$m$ coefficients

$$\mathbf{g}^m = \begin{pmatrix} g_{n'}^m \\ g_{n'+1}^m \\ g_{n'+2}^m \\ \vdots \\ g_N^m \end{pmatrix} = \begin{pmatrix} S_{n'+2}^m & S_{n'+3}^m & \cdots & S_N^m \\ \tilde{Q}_{n'+2}^m & \tilde{Q}_{n'+3}^m & \cdots & \tilde{Q}_N^m \\ 1 & 0 & \cdots & 0 \\ 0 & 1 & \cdots & 0 \\ \vdots & \vdots & \ddots & \vdots \\ 0 & 0 & \cdots & 1 \end{pmatrix} \begin{pmatrix} g_{n'+2}^m \\ g_{n'+3}^m \\ \vdots \\ g_N^m \end{pmatrix} = \mathbf{A}^m \mathbf{g}^{m'} \qquad (A6)$$

where the first two rows of the matrix $\mathbf{A}^m$ correspond to Equations A3 and A5, and the remaining rows comprise an identity matrix.

*Author contributions.* SMH conceived of the study, prepared the manuscript, compiled the measurement databases, produced all figures, and derived the model coefficients. HV and KML gave advice on model definition, study design, and definition of model outputs. KML also contributed the AMPS inversion code, which SMH adapted for deriving the Swarm Hi-C model. JPR helped with study design and presentation and wrote portions of the manuscript. JKB, LL, and DJK provided guidance on Swarm EFI measurements and provided feedback on model design and the manuscript. MM provided guidance on model definition and inverse theory and wrote part of the model description. HH carried out comparisons of the Swarm Hi-C and Swipe models with multiple existing models for validation, and provided feedback on the manuscript.





*Competing interests.*   The authors declare that they have no conflict of interest.

*Acknowledgements.*   This study is supported as part of Swarm Data, Innovation, and Science Cluster (DISC) activities, and is funded by ESA contract no. 4000109587/13/I-NB as well as the Trond Mohn Foundation, Research Council of Norway Contracts 300844 and 223252/F50. The work of HV and HT is supported by the Academy of Finland project 354521. JKB and LL were supported with funding from Canadian Space Agency Grant 21SUSTSHLE. KML was also funded by the European Union (ERC, DynaMIT, 101086985). Views and opinions

expressed are however those of the authors only and do not necessarily reflect those of the European Union or the European Research Council. Neither the European Union nor the granting authority can be held responsible for them.



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
