# Peer review of "Does high-latitude ionospheric electrodynamics exhibit hemispheric mirror symmetry?"

_EGUsphere, 2023_

## Referee Comment (RC2)

"Does high-latitude ionospheric electrodynamics exhibit hemispheric mirror
 symmetry?" by Spencer Mark Hatch, H. Vanhamäki, K.M. Laundal et al., 2023
==========================================================================

1. General Comments
===================
The paper presents comprehensive data sets of electrodynamic parameters
obtained by in-situ observations onboard nearly circular, polar-orbiting LEO
satellites in the upper ionosphere. The focus of this challenging study is
on the interhemispheric differences between mirror-symmetric patterns of
electrodynamic parameters obtained at high geomagnetic latitudes. The mirror
symmetry between hemispheres is defined here with respect to opposite signs
in both dipole tilt and IMF By component in GSM coordinates.

Novel empirical models of both ionospheric drift and currents are presented,
which are based on consistent building principles and measurements from
similar satellites: CHAMP and Swarm data for the empirical current's models
AMPS, and newly TII observations of the Swarm satellites for the high-latitude
ionospheric drift model called Hi-C.

Based on these empirical models, which are independently developed for both
hemispheres, a series of further patterns of various electrodynamic parameters
are revealed with their variations with respect to season (tilt angle) and IMF
clock angle in the By/Bz plane of GSM coordinates. These parameters comprise
the electromagnetic work as well as Hall and Petersen conductances, deduced
provided that the neutral wind at high latitudes can be considered as corotating
with the Earth.

The manuscript is of good quality, well written and organized and has a plenty
of Figures, which illustrate the physics of ionospheric electrodynamics and its
degree of interhemispheric symmetry optimally. The compilation of formulas is
also notably (37) and gives the manuscript partly the character of a good review
paper on empirical modeling.

2. Specific Comments
====================
The comparison between the cross-polar cap potential (CPCP) values of various
data sets with the empirical model Hi-C of the Swarm satellites in Fig. 5
is quite illuminating. The interhemispheric ratio of the Hi-C model is contrary
to both PRO2 and FH15, while the CS10 model corresponds partly to the present
study, though with a different clock angle dependence. For purely southward Bz,
the CS10 is in favour of the PR02 and FH15 models, while the NH/SH ratio for the
other IMF orientations is similar to the Hi-C findings.

The IMF dependence of the CS10 model is hence quite variable with respect to
the other data sets. I think that this is due to the finite latitudinal extent
of the CS10 SuperDARN network at that time. It would be of interest to know,
how the pattern of interhemispheric ratio looks like with the present-day,
more extented SuperDARN network. I'm not aware of any study in this regard.

The results of this Hi-C study suggest, that the PR02 and FH15 models might
feature some seasonal biases toward a preference of local winter patterns.
The highly elliptical orbit of the Cluster satellites might indeed have
generated some seasonal imbalance of the mapped high-latitude drift patterns.
This should be investigated independently. The short data set of PR02 was
taken near equinox and disagrees therefore somehow with the present findings.

In relation to Hall conductances (section 5.2, page 25), just at one place
in line 469, the standard deviation of Hall conductances are mentioned. I
wonder, where this pattern is shown in the manuscript or in some supplement
materials? Unless I'm very much mistaken, standard deviations of any parameters
are not shown nor discussed elsewere in the manuscipt.

The notion regarding the role of neutral winds on line 532ff. is remarkable in my mind. However, the criteria given in Equation 37 alone is not sufficient to determine the locations, where the assumption that the neutral gas simply corotates with the Earth breakes down. I'm missing here a somehow better identification of those circumstances (IMF orientation, season), where "... namely negative or unphysically large conductances and sharp conductance gradients..." (line 538) occur.

3. Technical Corrections
========================
The manuscript is very well written with almost no misprints (the very few exceptions that I found are listed at the bottom). The Figures, however, could still be somehow improved in my mind (see remarks below).

In Section 5.2 (Hall conductance) and 5.3 (Petersen conductance), reference is made to Supporting Information of Figs. (S1)-(S6), which I couldn't find in the Preprint community platform.

I like the idea of direct interhemispheric comparisons in one and the same plot by using isolines and colored contours simultaneously for the opposite hemispheres. A problem might arise from the fact that the Figure's legends in Figs. 2-4 and 6-8 show a continuous color bar, while the contours are discrete. This is made differently for Figs. 10-15.

Yes, the potential range for the various panels in the Figs. 2-4 is quite dynamic and therefore also the potential steps are quite variable. I agree that it is reasonable to keep the number of contours and contour lines small. However, it might be useful to indicate the common constant step sizes then for each panel individually within the inscription blocks.

The inlets (or inscription blocks) of Figs. 2-4 and 6-8 provide parameter values of the Northern and Southern hemisphere with 1-2 digits, while the ratio of the value is given with three digits. This allows some space for speculations about the correct numbers as, e.g., for the upper left panel of Fig.7 with values for $W_N$ and $W_S$ between 6.4 and 7.6 GW. I think it would be better to provide about the same number of digits for the parameter values as for their ratio.

Line 197: just below eq. (15) after "with d1 and d2" I miss a verb or "as"
Line 438: parenthesis for the reference not needed here
Line 461: "Bz" is probably meant here instead of "By"
Line 585: one "in" should be deleted
Line 618: "HH" is probably "HT"(?)

---

## Referee Comment (RC3)

The paper *Does high-latitude ionospheric electrodynamics exhibit hemispheric mirror symmetry?* provides a comprehensive perspective over high-latitude ionospheric electrodynamics based on almost 9 years of Swarm data. The paper brings a significant contribution to the field and is certainly suitable for publication. Before that, however, perhaps there is room to optimize the transmission of the paper's message to the reader.

The (very) comprehensive character of the paper is both a merit and also an issue. Its substance could easily fill three papers, I would say, focused as follows and, piecewise, easier to absorb:
- One paper on the Hi-C convection model, perhaps including more details on the math (Section 3) in a less dense presentation;
- Another paper on combining Hi-C with AMPS to produce Swipe, and the resulting maps on electromagnetic work and conductance, $W$, $\Sigma_P$, $\Sigma_H$. This paper could also benefit from more discussion of the results.
- The (a)symmetry between northern and southern hemisphere could be the object of yet another paper, once again assisted possibly by more discussion.

As of now, the (a)symmetry 'paper' also gives the title of the full manuscript, while the visibility of the other two 'papers' is somewhat obscured. This might be regarded as a weakness, even if a rather uncommon one.

In the following I make a few comments on each of these three 'papers', then list a few more issues that may require the authors' attention. More comments on Hi-C, which feels also the more demanding.

**1. The Hi-C 'paper'**

a) The derivation of 2D maps from the 1D cross-track TII measurements may deserve some discussion beyond the math. To some extent, this has been done also before, by Lomidze et al. (2019), and was validated by comparison with Weimer (2005). However, Lomidze et al. (2019) concentrated on the cross-track component, whereas here both components of the convection are derived. My understanding is that the 'jump' from 1D individual measurements to 2D statistical results is related essentially to the potential nature of the electric field, that prevails most of the time, and then convection is dominated by electric drift. But I wonder if the 1D character of the measurement does not still have some impact, in particular on the accuracy / error margin of the results (see also 4e). While the error margin is beyond the scope of the paper (see also point 3), it may still be worth to comment on this matter. For example, when the convection map shows plasma velocity mainly in E-W, cross-track direction, I expect this result is more accurate than plasma velocity shown by convection map mainly in N-S, along-track direction. To some extent, this reminds me the SuperDARN maps, where the model is based on solar wind parameters, though one expects better accuracy around the radar measurement points.

b) The Modified Apex coordinates, MA-110, appear to play an important role in the formalism, taking care, e.g., of the mapping from Swarm altitude to ionosphere (L180–181, L221–227) or the distortions of the magnetic field (L193–194, L575–576). For details regarding MA-110 the reader is referred to the paper by Richmond (1995). While this is in principle fine, it would be good to provide more clarifications, starting with the definition of the apex altitude (L202). Other features that could be detailed a bit are the way MA-110 takes care of i) mapping and ii) magnetic field distortions (mentioned above), iii) the possible bias of the SH model (L190–194), iv) how strongly non-orthogonal are ($\mathbf{d}_1$, $\mathbf{d}_2$ – L197) and ($\mathbf{e}_1$, $\mathbf{e}_2$ – L207) (both pointing roughly in the same directions), v) what is roughly the difference between the two and the standard spherical system (L223–224) (the energy argument at L237 suggests perhaps less than 1°?), vi) some brief explanation of the difference between 44° QD latitude (L166) and 47° MA-110 (L250, L256) (is 47° MA-110 just the average of 44° and 50° QD, L251? how significant is actually the difference between QD and MA-110?).

c) The decimation at L168-169 is not explained (together with the related limitation to scales larger than some 40 km). Is this because of further matching with AMPS?

d) The two paras at the end of Section 6, L556-569, are quite helpful to understand the math, but they might fit better to Section 3, where the math is done (perhaps in Section 3.2?). Those paras are quite specific, they do not seem to belong to the Discussion section.

**2. The Swipe 'paper'**

a) I think an important issue here is the error margin. While the authors eliminate problematic regions by asking the Hall and Pedersen conductance to be positive (with more constraint on Pedersen, via the threshold of 0.5 mW/m^2 in Eq. 37), the error margin is deferred to another study (L383–386). Nonetheless, a rough estimate of the uncertainty of $W_N$ and $W_S$ is provided at L436-437. Could this estimate be briefly explained? And perhaps similar estimates could be provided and tentatively discussed for other quantities, like the conductances? Based on 'common sense' knowledge, conductances around a couple of mho and less are more and more uncertain, the lower the conductance is. On the other hand, low conductance areas are actually quite broad and can make a significant contribution to Joule heating, which is a major driver of ionospheric electrodynamics. More discussion on the error margin seems appropriate, even if the actual (quantitative) solution is beyond the scope.

b) Related to this matter, the threshold of 0.5 mW/m^2 for W is explained at L379–382 based on typical values of the constituents – neutral velocity, magnetic field, and sheet current – together with a qualitative remark on apparition of *very sharp gradients* in the conductance below this threshold. Given the importance of this threshold for the low conductance areas and for the validation of the results (L138–139), it would be nice to elaborate a bit and quantify roughly the *very sharp gradients*.

**3. The (a)symmetry 'paper'**

a) In the literature one can find two different perspectives on the (a)symmetry between northern and southern hemispheres: Papers of the sort here, that look at the northern and southern hemisphere under similar conditions of tilt angle and IMF By, as well as papers that concentrate on the instantaneous asymmetry – driven, to a large extent, by the different conductance between the summer and winter hemisphere (and also by the tilt angle and By, whose values are not mirrored for such studies). Judging just by title, one could question what is the perspective here, before the matter becomes clear in the text. It may still be worth to comment a bit on these two complementary facets of the (a)symmetry.

**4. Other issues**

a) L153–154: This sentence is correct, but the association between the neutral wind and the definition of the Poynting flux (Eq. 14), rooted in the energy conservation Poynting theorem, may drive some confusion.

b) L155: Perhaps *Methodology and data for Hi-C*? This is, indeed, the core, but methodology includes also combination with AMPS, and then exploring (a)symmetry in all quantities (the three 'papers'…).

c) L156 – 158: Perhaps say simply that \hat{y} = \hat{x} \times \hat{r} / |\hat{x} \times \hat{r}|?

d) Eq. 22: The tilt angle is not under sin or cos, like the clock angle. Is this because the tilt angle is (rather) small?

e) L304–305: Please explain briefly the origin of the ill-condition. Can this be related, at least to some extent, to the 1D measurements (point 2a above)?

f) L312–313: Not sure I understand: sectorial resolution, associated with M, is not the same with zonal (or longitudinal)? And N is not associated with latitudinal resolution, rather than zonal?

g) L372: Could you describe / illustrate briefly the differences between Hi-C and AMPS?

h) Eqs. 37 indicate a difference in the treatment of $\Sigma_P$ and $\Sigma_H$, in that for $\Sigma_P$ some margin is considered above zero (via the 0.5 mW/m$^2$ threshold). Please comment a bit on this difference, how comes that no margin is needed for $\Sigma_H$?

i) L392–394: This appears to hold in particular for local winter.

j) L414–415: *neutral wind field corotating with the Earth* is a bit confusing. Strictly, this means no neutral wind, whereas what is likely meant is that the neutral wind has the same direction as the Earth rotation.

k) L423–424: Any comment on the dominant direction of the neutral wind? (considering also the previous para)

l) L451 and L480: The Supporting Information is missing (Figures S1–S6).

m) L484–486: This suggests that hot spots might be related to IMF By (?).

**5. Typos and alike**

L101 and L129: Define the LHS and RHS acronyms; L181 and Fig. 1: The black lines are hardly visible, change black to some color (?). Add scales to the distributions (?); L208: Laundal et al. (2018) useD; L224, L225: EquationS 16, 18; L312: sectorial, lower than than; L341: If the average CPCP is 51 kV for both NH and SH, it cannot be 3% greater in the SH, as stated in the caption of Fig. 2; L355: similar -> more or less similar (?); L362: contours -> colored contours (?); Figs. 6–8: The black line and color contours can be compared easily in terms of shape, less easily in terms of value. The integral values at the top right corners help; L409–410: Delete from 'as mentioned…' to the end of the sentence. L439: Regardless of $B_T$ (?); L460: By -> IMF By; L461: IMF By -> IMF Bz; L474: distribution OF; L516: examines (?), mirror asymmetry -> mirror symmetry; L522: These studies are of relevance to this study; L537-538: Weimer and Edwards (2021) -> WE21; L584: field field; L618: HH -> HT (?).

---

## Author Comment (AC3)

**Response to Reviewer #3 (Octav Marghitu)**

RC3: 'Comment on egusphere-2023-2920', Octav Marghitu, 17 Jan 2024

We thank the reviewer for their careful reading of our manuscript and their comments and suggestions. Below the reviewer's comments are shown in **black**, and our responses are shown in **blue**.

The paper Does high-latitude ionospheric electrodynamics exhibit hemispheric mirror symmetry? provides a comprehensive perspective over high-latitude ionospheric electrodynamics based on almost 9 years of Swarm data. The paper brings a significant contribution to the field and is certainly suitable for publication. Before that, however, perhaps there is room to optimize the transmission of the paper's message to the reader.

The (very) comprehensive character of the paper is both a merit and also an issue. Its substance could easily fill three papers, I would say, focused as follows and, piecewise, easier to absorb:

• One paper on the Hi-C convection model, perhaps including more details on the math (Section 3) in a less dense presentation;

• Another paper on combining Hi-C with AMPS to produce Swipe, and the resulting maps on electromagnetic work and conductance, $W$, $\Sigma_P$, $\Sigma_H$. This paper could also benefit from more discussion of the results.

• The (a)symmetry between northern and southern hemisphere could be the object of yet another paper, once again assisted possibly by more discussion.

As of now, the (a)symmetry 'paper' also gives the title of the full manuscript, while the visibility of the other two 'papers' is somewhat obscured. This might be regarded as a weakness, even if a rather uncommon one.

- We agree with the reviewer that this study is outside the norm in terms of length (also indirectly hinted at by the second reviewer), and we agree that our choosing to squeeze all three "papers" into a single study is not unproblematic. For us, this choice came down to a simple question of available bandwidth; we found it most expedient to write a single and perhaps unfortunately lengthy paper.

  We have attempted to partially make up for this weakness by making the model visible (announced so far on three different mailing lists), open source, and easily accessible (available on PyPI, GitHub, Swarm VirES, and Zenodo).

In the following I make a few comments on each of these three 'papers', then list a few more issues that may require the authors' attention. More comments on Hi-C, which feels also the more demanding.

1. The Hi-C 'paper'

a) The derivation of 2D maps from the 1D cross-track TII measurements may deserve some discussion beyond the math. To some extent, this has been done also before, by Lomidze et al. (2019), and was validated by comparison with Weimer (2005). However, Lomidze et al. (2019) concentrated on the cross-track component, whereas here both components of the convection are derived. My understanding

is that the 'jump' from 1D individual measurements to 2D statistical results is related essentially to the potential nature of the electric field, that prevails most of the time, and then convection is dominated by electric drift. But I wonder if the 1D character of the measurement does not still have some impact, in particular on the accuracy / error margin of the results (see also 4e). While the error margin is beyond the scope of the paper (see also point 3), it may still be worth to comment on this matter. For example, when the convection map shows plasma velocity mainly in E-W, cross-track direction, I expect this result is more accurate than plasma velocity shown by convection map mainly in N-S, along-track direction. To some extent, this reminds me the SuperDARN maps, where the model is based on solar wind parameters, though one expects better accuracy around the radar measurement points.

- We confirm the reviewer's understanding that the jump from 1D cross-track measurements to a derived 2D convection pattern relies heavily on the (assumed) potential nature of the electric field. We completely agree with the reviewer that this is an issue; it is the reason for the statement made on Line 255 of the original manuscript, where we state that "model convection in the north-south direction is … essentially unconstrained by measurements."

  In the revised manuscript we will insert a new paragraph between the first and second paragraphs of the discussion section. Here we will cite the previous validation work done by Lomidze et al (2019) via comparison with Weimer (2005), and make the reviewer's point that that validation effort concentrated on the cross-track component. We will then point out that we have not explicitly validated the model's predicted along-track convection, that the Swarm orbits are such that the direction of the cross-track component does not vary much from orbit to orbit, and that the uncertainty in the along-track direction must of necessity be larger since it is not constrained by measurements. We will also refer to Lomidze et al (2021), Section 4.2.2, where they elaborate on the data quality of the along-track ion drift "proxy", and provide a clear rationale for excluding along-track ion drift.

b) The Modified Apex coordinates, MA-110, appear to play an important role in the formalism, taking care, e.g., of the mapping from Swarm altitude to ionosphere (L180–181, L221–227) or the distortions of the magnetic field (L193–194, L575–576). For details regarding MA-110 the reader is referred to the paper by Richmond (1995). While this is in principle fine, it would be good to provide more clarifications, starting with the definition of the apex altitude (L202). Other features that could be detailed a bit are the way MA-110 takes care of i) mapping and ii) magnetic field distortions (mentioned above), iii) the possible bias of the SH model (L190–194), iv) how strongly non- orthogonal are ($d_1$, $d_2$ – L197) and ($e_1$, $e_2$ – L207) (both pointing roughly in the same directions), v) what is roughly the difference between the two and the standard spherical system (L223–224) (the energy argument at L237 suggests perhaps less than 1°?), vi) some brief explanation of the difference between 44° QD latitude (L166) and 47° MA-110 (L250, L256) (is 47° MA-110 just the average of 44° and 50° QD, L251? how significant is actually the difference between QD and MA-110?).

- We thank the reviewer for pointing out that many of these points simply were not explained in enough detail. In the revised manuscript we will explain that the representation of the electric field and convection coefficients in Equations 16 and 18, respectively, explicitly indicate that these coefficients are constant along field lines, regardless of whether the field lines themselves depart from dipolarity (we know that they do). We will also include some general comments about how strongly nonorthogonal the various basis vectors are (poleward of ±60° MA-110 latitude, the angles between $d_1$ and $d_2$ and $e_1$ and $e_2$ do not deviate from orthogonality by more than 15° in either hemisphere), and refer the reader to the review paper by Laundal and Richmond (2016), which

explores many of these and related questions.

Regarding the difference between QD and MA-110 coordinates, QD coordinates are not strictly constant along field lines, but MA coordinates (including MA-110 coordinates) are. So QD coordinates are good for things that vary with height, such as the B-field of divergence-free currents, while MA coordinates are good for things that map along field lines, like E-fields, convection velocities, and FACs.

 Laundal and Richmond (2016) also describe that in practice, the difference is essentially the reference height to which the dipole mapping is done. In QD coordinates, the reference height is the height of the point in question. In MA-110 coordinates, the reference height is 110 km. This means that if the point in question is at $h$=110 km, QD coordinates and MA-110 coordinates are identical. For points above this height, the QD latitude is equatorward of the MA-110 latitude, and vice versa for points below this height. (See Equation 43 in Laundal and Richmond, 2016). At Swarm altitudes of ~400–500 km, an MA-110 latitude of 47° corresponds to QD latitudes of ~45.4°–45.8°.

In the revised manuscript we will add a succinct summary of these additional points to the existing discussion, and make a clearer reference to the other documents where detailed information can be found, including the previously cited Richmond (1995) paper as well as Laundal and Richmond (2016, doi: 10.1007/s11214-016-0275-y), Emmert et al (2010, doi: 10.1029/2010JA015326), and Laundal et al (2016, doi: 10.1186/s40623-016-0518-x).

c) The decimation at L168-169 is not explained (together with the related limitation to scales larger than some 40 km). Is this because of further matching with AMPS?

- In the revised manuscript we will point out that this choice was made because we found that increasing the effective measurement cadence (i.e., including more measurements) did not visibly affect the shape of the potential patterns.

d) The two paras at the end of Section 6, L556-569, are quite helpful to understand the math, but they might fit better to Section 3, where the math is done (perhaps in Section 3.2?). Those paras are quite specific, they do not seem to belong to the Discussion section.

- This is a fair point, in the revised manuscript we will move these paragraphs to Section 3.2.

2. The Swipe 'paper'

a) I think an important issue here is the error margin. While the authors eliminate problematic regions by asking the Hall and Pedersen conductance to be positive (with more constraint on Pedersen, via the threshold of 0.5 mW/m^2 in Eq. 37), the error margin is deferred to another study (L383–386). Nonetheless, a rough estimate of the uncertainty of $W_N$ and $W_S$ is provided at L436-437.  Could this estimate be briefly explained? And perhaps similar estimates could be provided and tentatively discussed for other quantities, like the conductances? Based on 'common sense' knowledge, conductances around a couple of mho and less are more and more uncertain, the lower the conductance is. On the other hand, low conductance areas are actually quite broad and can make a significant contribution to Joule heating,

which is a major driver of ionospheric electrodynamics.  More discussion on the error margin seems appropriate, even if the actual (quantitative) solution is beyond the scope.

- We agree that the question of uncertainty is one that deserves serious attention, and as the reviewer notes, we think this is most properly addressed as part of a dedicated study. To our knowledge, no existing study involving development and demonstration of an empirical model of quantities such as ionospheric potential has yet taken up the question of uncertainty estimation.

  We agree that the statement that we made in the original manuscript was too vague to be of much use. In the revised manuscript, we will refer the reader to a second, very brief appendix where we state how one may use standard error propagation techniques together order-of-magnitude estimates of the uncertainties of the magnitudes of J, E, and the angle between them to arrive at a very rough estimate uncertainty of the hemispheric integrated EM work that turns out to be of order 0.1–1 GW.

  Regarding the uncertainty of the conductance: Since the conductance depends critically on having a proper estimate of the neutral winds and is generally much more poorly understood than the electromagnetic work and/or Joule heating, we feel it is best to defer discussion of the uncertainty of conductance estimates to a dedicated study. We hope the reviewer will understand that our desire is to do the subject justice, and we do not feel this is possible within the present manuscript given the (already large) scope of the study.

b) Related to this matter, the threshold of 0.5 mW/m^2 for W is explained at L379–382 based on typical values of the constituents – neutral velocity, magnetic field, and sheet current – together with a qualitative remark on apparition of very sharp gradients in the conductance below this threshold.  Given the importance of this threshold for the low conductance areas and for the validation of the results (L138–139), it would be nice to elaborate a bit and quantify roughly the very sharp gradients.

- Rather than adding more text, we propose to add a note in the vicinity of the lines indicated by the reviewer where we state that we include a Supporting Information figure that shows typical distributions of the conductances when no screening criteria are applied, such as the following:

[Figure]

Here one can see, for example, locations indicated by black boxes where the conductances change by as much as 50 mho from one grid cell to the next. There are also many cells within which the conductance is negative.

Of course, the locations of the sharp gradients and negative conductances are not static, but vary with IMF orientation, solar wind speed, and such. We have not spent any significant amount of time trying to identify where the negative conductances or sharp gradients occur, since we know that our model is missing key information about the neutral winds.

3. The (a)symmetry 'paper'

a) In the literature one can find two different perspectives on the (a)symmetry between northern and southern hemispheres: Papers of the sort here, that look at the northern and southern hemisphere under similar conditions of tilt angle and IMF By, as well as papers that concentrate on the instantaneous asymmetry – driven, to a large extent, by the different conductance between the summer and winter hemisphere (and also by the tilt angle and By, whose values are not mirrored for such studies). Judging just by title, one could question what is the perspective here, before the matter becomes clear in the text. It may still be worth to comment a bit on these two complementary facets of the (a)symmetry.

- We appreciate the reviewer's making this distinction between the types of studies that one encounters in the literature. In the revised manuscript we propose to make the following addition to the discussion, which relies heavily on the language that the reviewer has used (we thank the reviewer and will acknowledge their contribution in the acknowledgements):

  In the literature one encounters different approaches to the topic of symmetry between the two hemispheres that may be roughly separated into two categories: those that examine asymmetries in the NH and SH under complementary conditions of tilt angle and IMF By,

and those that concentrate on instantaneous asymmetries that are driven, to a large extent, by differences in conductance between the summer and winter hemispheres (but also by tilt angle $\psi$ and IMF $B_y$, whose values are not mirrored). This study belongs to the former category.

4. Other issues

a) L153–154: This sentence is correct, but the association between the neutral wind and the definition of the Poynting flux (Eq. 14), rooted in the energy conservation Poynting theorem, may drive some confusion.

- Good point, in the revised manuscript we will make the following change:

  The neutral wind $\mathbf{v}_n$ notably does not appear in Equation 14, as the Poynting flux is frame dependent and arises in connection with the well-known energy conservation (Poynting) theorem.

b) L155: Perhaps Methodology and data for Hi-C? This is, indeed, the core, but methodology includes also combination with AMPS, and then exploring (a)symmetry in all quantities (the three 'papers'…).

- In the revised manuscript we will change the name of this section to "Methodology and data for Swarm Hi-C model" as suggested

c) L156 – 158: Perhaps say simply that \hat{y} = \hat{x} \times \hat{r} / |\hat{x} \times \hat{r}|?

- Thank you for the suggestion, we will revise the manuscript accordingly.

d) Eq. 22: The tilt angle is not under sin or cos, like the clock angle. Is this because the tilt angle is (rather) small?

- Precisely, this is because the tilt angle remains within a (relatively) narrow range of ~±30°.

e) L304–305: Please explain briefly the origin of the ill-condition. Can this be related, at least to some extent, to the 1D measurements (point 2a above)?

- As acknowledged above, the fact that the measurements are 1D must have some influence on the derivation of the model. However, it is unlikely that the need for regularization is connected with the fact that the measurements are 1D. Without regularization the cost function would be entirely dependent on measurement-model misfit (first term on RHS of Equation 35). With such a simple cost function the norm of the model parameter vector $\mathbf{k}$ generally ends up being far too large, corresponding to overfitting or in the worst case numeric overflow.

f) L312–313: Not sure I understand: sectorial resolution, associated with M, is not the same with zonal (or longitudinal)? And N is not associated with latitudinal resolution, rather than zonal?

- We apologize for the possibly confusing choice of words here. In the revised manuscript we will write "That $M$ is much less than $N$ indicates that the longitudinal resolution of the model is much lower than than the latitudinal resolution."

g) L372: Could you describe / illustrate briefly the differences between Hi-C and AMPS?

- The difference between the models that we had in mind, which is implicitly referred to here but not explicitly stated, is that the perpendicular current vector **J** from AMPS and the electric field vector **E** from Swarm Hi-C are not in any way "co"-constrained to ensure that **J**·**E** > 0 and (**J**x**E**)·**b** > 0, as required by the physics (NOTE: assuming the neutral wind is zero in the Earth's rotating frame of reference!). In the absence of information about the neutral wind, it seems impossible to make a meaningful comment about how the models might be different if they had been co-constrained.

h) Eqs. 37 indicate a difference in the treatment of $\Sigma_P$ and $\Sigma_H$, in that for $\Sigma_P$ some margin is considered above zero (via the 0.5 mW/m 2 threshold). Please comment a bit on this difference, how comes that no margin is needed for $\Sigma_H$ ?

- The reason for imposing a different threshold is that we found the two thresholds correspond to different types of issues with the conductance distributions, as discussed on Lines 376–378: the 0.5 mW/m² criterion primarily addresses the issue of negative or unphysically large conductances or large conductance gradients within the polar cap and equatorward of ±60° MLat, while the $\Sigma_H$ > 0 mho requirement mostly addresses locations poleward of 70° where the Hall conductances may be negative (between -6 and -1 mho for the conditions we examined).

  In other words, these values were chosen on a heuristic basis, and we by no means intend to imply that they are somehow the "right" values (please see our comment to the second reviewer on this point). The simple fact of the matter is that for the conditions we examined, when we reduced the first threshold to 0 mW/m² or even 0.3 mW/m², we found that the Pedersen and Hall conductance distributions still evinced sharp gradients and/or negative or extremely large values. We chose the (approximate) minimum value necessary to mask regions with these problems, and found that this minimum value corresponds to what might be considered the typical contribution from a height-averaged neutral wind with a magnitude of 100 m/s and a current sheet density of 100 mA/m, as described on lines 379–382.

  We will add some explanatory remarks along these lines below Equation 37 in the revised manuscript.

i) L392–394: This appears to hold in particular for local winter.

- Thank you for your careful examination of these figures, we will add this comment on the relevant lines in the revised manuscript.

j) L414–415: neutral wind field corotating with the Earth is a bit confusing. Strictly, this means no neutral wind, whereas what is likely meant is that the neutral wind has the same direction as the Earth rotation.

- This is of course a question of frame of reference, such that if one is standing on Earth we do indeed mean $\mathbf{v}_n$ = 0 (no neutral wind), whereas if one is looking down at the Earth in, say, a GSM frame of reference the neutral wind field rotates together with Earth.

  We thank the referee for pointing out that this may be a source of confusion, and in the revised manuscript we will change the phrase "reducing the electric field in the neutral wind frame" on Line 416 so that it reads "reducing the electric field in Earth's rotating frame of reference".

k) L423–424: Any comment on the dominant direction of the neutral wind? (considering also the previous para)

- We agree that this would be valuable to comment upon. Unfortunately we do not feel comfortable commenting on the dominant direction of the neutral wind, as the presence of neutral wind shears at the relevant altitudes greatly complicate determination of a dominant direction. As mentioned on Line 125, this is a topic that we are actively researching.

l) L451 and L480: The Supporting Information is missing (Figures S1–S6).

- Thank you (and the other reviewers) for catching this oversight on our part. We will be sure to include them during resubmission.

m) L484–486: This suggests that hot spots might be related to IMF By (?).

- This is a good point that we will include on this line in the revised manuscript.

5. Typos and alike

L101 and L129: Define the LHS and RHS acronyms;

L181 and Fig. 1: The black lines are hardly visible, change black to some color (?). Add scales to the distributions (?);

L208: Laundal et al. (2018) useD;

L224, L225: EquationS 16, 18;

L312: sectorial, lower than than;

L341: If the average CPCP is 51 kV for both NH and SH, it cannot be 3\% greater in the SH, as stated in the caption of Fig. 2; L355: similar -> more or less similar (?);

L362: contours -> colored contours (?);

Figs. 6–8: The black line and color contours can be compared easily in terms of shape, less easily in terms of value. The integral values at the top right corners help;

L409–410: Delete from 'as mentioned…' to the end of the sentence.

L439: Regardless of B T (?);

L460: By -> IMF By;

L461: IMF By -> IMF Bz;

L474: distribution OF;

L516: examines (?), mirror asymmetry -> mirror symmetry;

L522: These studies are of relevance to this study;

L537-538: Weimer and Edwards (2021) -> WE21;

L584: field field;

L618: HH -> HT (?).

- Thank you for catching all of these typos and small mistakes. We will correct all of them according to the referee's suggestion in the revised manuscript.

---

## Author Response (AR1)

**Response to Reviewer #1 (Daniel Weimer)**

RC1: 'Comment on egusphere-2023-2920', Daniel Weimer, 03 Jan 2024

https://doi.org/10.5194/egusphere-2023-2920-RC1

We thank the reviewer for their critical reading of and commentary on our manuscript. Below the reviewer's comments are shown in **black**, and our responses are shown in **blue**.

This paper reports the results of a study of high-latitude ionospheric electrodynamics in both the Northern and Southern hemispheres to examine whether or not the patterns have mirror symmetry under reversal of sign changes in dipole tilt angle and IMF B_y component. It employs a model of the electric fields developed from newly available Swarm electric field measurements (thru spring of 2023) combined with the existing AMPS models of magnetic perturbations and currents. The results show that, with minor differences, the mirror symmetry holds. Even more significant is that maps of the ionospheric Hall and Pedersen conductances are derived.

The writing is of excellent quality, with no errors detected. One major fault , easily corrected, is that Supporting Information (SI) with figures S1 through S6, that are mentioned in the text, are not provided.

- Thank you (and the other reviewers) for pointing out that we failed to include these figures, we have now included them with the revised manuscript.

The important conductance results don't seem to be given the attention that is deserved in the title and abstract. This paper might be missed by some researchers who are interested in these conductance values. It also would be useful if there was a way to distribute maps of the conductances in numerical form for a wide range of solar wind, IMF, and tilt angles.

- We agree with the referee that the conductance maps are likely the most important result of this paper.

  We have attempted to make conductance values easy to access by releasing an open-source front-end for the Swipe model (written in Python) complete with examples and code to reproduce all figures shown in the manuscript. Additionally, encouraged by this comment from the reviewer, we have announced the release of the model on many of our community's mailing lists (AGU SPA, CEDAR, and European Heliophysics). We nevertheless agree with the reviewer that it would be ideal to find a way to make the conductance maps even more accessible, and we are at present working on developing a web interface that would allow one to easily export conductance values to an ASCII or .csv file.

  For now, we wish to point out that the Swipe model may be run directly from ESA's VirES platform without needing to install or run Python on one's machine. (Making an account is free, and examples of running the Swipe model are given here: https://notebooks.vires.services/notebooks/07c1_sw-pyswipe )

While it may be beyond the scope of this paper (and page limits) it would be useful to see how the models perform for IMF magnitudes up to 15 nT and higher, even though there is little IMF data in that range.

- This is a very good suggestion. One of our contractual obligations with ESA is to produce a "validation report" of the Swipe model consisting of a detailed comparison with many other models, including the Weimer model. This validation report addresses the very large storm that was the subject of the Rastätter et al (2016) Poynting flux challenge paper, and shows how the Swipe model performs for large IMF magnitudes. The results are very interesting in our opinion, but are too large for inclusion as

part of the peer-reviewed manuscript. We have therefore included this validation report as Supporting Information for the manuscript (it is also available from ESA: (https://earth.esa.int/eogateway/documents/d/earth-online/swarm-swipe-validation-report ).

**Response to Reviewer #2 (Matthias Förster)**

RC2: 'Comment on egusphere-2023-2920', Matthias Förster, 09 Jan 2024

We thank the reviewer for their careful reading of our manuscript and their comments and suggestions. Below the reviewer's comments are shown in **black**, and our responses are shown in **blue**.

**1. General Comments**
=====================

The paper presents comprehensive data sets of electrodynamic parameters obtained by in-situ observations onboard nearly circular, polar-orbiting LEO satellites in the upper ionosphere. The focus of this challenging study is on the interhemispheric differences between mirror-symmetric patterns of electrodynamic parameters obtained at high geomagnetic latitudes. The mirror symmetry between hemispheres is defined here with respect to opposite signs in both dipole tilt and IMF By component in GSM coordinates.

Novel empirical models of both ionospheric drift and currents are presented, which are based on consistent building principles and measurements from similar satellites: CHAMP and Swarm data for the empirical current's models AMPS, and newly TII observations of the Swarm satellites for the high-latitude ionospheric drift model called Hi-C.

Based on these empirical models, which are independently developed for both hemispheres, a series of further patterns of various electrodynamic parameters are revealed with their variations with respect to season (tilt angle) and IMF clock angle in the By/Bz plane of GSM coordinates. These parameters comprise the electromagnetic work as well as Hall and Petersen conductances, deduced provided that the neutral wind at high latitudes can be considered as corotating with the Earth.

The manuscript is of good quality, well written and organized and has a plenty of Figures, which illustrate the physics of ionospheric electrodynamics and its degree of interhemispheric symmetry optimally. The compilation of formulas is also notably (37) and gives the manuscript partly the character of a good review paper on empirical modeling.

**2. Specific Comments**
=====================

The comparison between the cross-polar cap potential (CPCP) values of various data sets with the empirical model Hi-C of the Swarm satellites in Fig. 5 is quite illuminating. The interhemispheric ratio of the Hi-C model is contrary to both PRO2 and FH15, while the CS10 model corresponds partly to the present study, though with a different clock angle dependence. For purely southward Bz, the CS10 is in favour of the PR02 and FH15 models, while the NH/SH ratio for the other IMF orientations is similar to the Hi-C findings.

The IMF dependence of the CS10 model is hence quite variable with respect to the other data sets. I think that this is due to the finite latitudinal extent of the CS10 SuperDARN network at that time. It would be of interest to know, how the pattern of interhemispheric ratio looks like with the present-day, more extented SuperDARN network. I'm not aware of any study in this regard.

- We agree that this would be interesting to know. We, like the reviewer, are not aware of any recent SuperDARN-based study that examines interhemispheric CPCP ratios.

The results of this Hi-C study suggest, that the PR02 and FH15 models might feature some seasonal biases toward a preference of local winter patterns. The highly elliptical orbit of the Cluster satellites might indeed have generated some seasonal imbalance of the mapped high-latitude drift patterns. This should be investigated independently. The short data set of PR02 was taken near equinox and disagrees therefore somehow with the present findings.

- We thank the reviewer for the very careful comparison with previous studies. On lines 380–386 of the revised manuscript we have included the observations made by the reviewer here.

In relation to Hall conductances (section 5.2, page 25), just at one place in line 469, the standard deviation of Hall conductances are mentioned. I wonder, where this pattern is shown in the manuscript or in some supplement materials? Unless I'm very much mistaken, standard deviations of any parameters are not shown nor discussed elsewere in the manuscipt.

- We realize that we did not make clear what was done here, thank you for pointing this out. In the revised manuscript have revised the text on lines 498–500 as follows to clarify this point:

  > The standard deviation of Hall conductances for these $\theta_c$ orientations within this same region (18–6 MLT and -75° to -60° MLat) is likewise lowest during local winter and highest during local summer. We obtain the standard deviation by first calculating the Hall conductances at points on a spherical grid with spacing of approximately 0.24 MLT and 0.3° MLat; the standard deviation is then calculated from all points within this region at which the criteria (37) are met.

The notion regarding the role of neutral winds on line 532ff. is remarkable in my mind. However, the criteria given in Equation 37 alone is not sufficient to determine the locations, where the assumption that the neutral gas simply corotates with the Earth breaks down. I'm missing here a somehow better identification of those circumstances (IMF orientation, season), where "... namely negative or unphysically large conductances and sharp conductance gradients..." (line 538) occur.

- This is a good point, and we agree that the criteria in Equation 37 are not entirely satisfactory. In light of the reviewer's comment we have revised the manuscript on lines 592–598 as follows:

  > These enforce the basic physical requirement that the height-integrated conductances be positive (note however that in a dusty plasma the Hall conductance may be negative; see, e.g., Shebanits et al., 2020, and references therein). They have nevertheless arisen heuristically in the course of this study as a means of screening out the negative or unphysically large conductances and sharp conductance gradients that otherwise appear. It would be greatly preferable to enforce positive conductances (i.e., physical consistency) as part of the model design, and to include relevant neutral wind measurements. Such improvements deserve attention in future studies.

3. Technical Corrections
========================

The manuscript is very well written with almost no misprints (the very few exceptions that I found are listed at the bottom). The Figures, however, could still be somehow improved in my mind (see remarks below).

In Section 5.2 (Hall conductance) and 5.3 (Pedersen conductance), reference is made to Supporting Information of Figs. (S1)-(S6), which I couldn't find in the Preprint community platform.

- We thank the reviewer (and the other reviewers) for pointing out that we failed to include these during submission. We have included them as part of our resubmission.

I like the idea of direct interhemispheric comparisons in one and the same plot by using isolines and colored contours simultaneously for the opposite hemispheres. A problem might arise from the fact that the Figure's legends in Figs. 2-4 and 6-8 show a continuous color bar, while the contours are discrete. This is made differently for Figs. 10-15.

- We chose to use continuous color bars for Figures 2–4 and 6–8 to allow for different contour spacing from panel to panel. Since the contour spacing is the same for all panels in Figures 10–15, we decided in these figures to use a discrete colorbar that indicates the contour spacing. We hope that the reviewer agrees that this was a sensible strategy.

  To make this clear for the reader, we have added the following text to the captions of Figures 10 and 13 in the revised manuscript:

  > In contrast to Figures 2–4 and 6–8, in this figure the contour spacing indicated by the colorbar is identical for all panels.

Yes, the potential range for the various panels in the Figs. 2-4 is quite dynamic and therefore also the potential steps are quite variable.  I agree that it is reasonable to keep the number of contours and contour lines small. However, it might be useful to indicate the common constant step sizes then for each panel individually within the inscription blocks.

- This is a good idea; we have modified Figures 2–4 and 6–8 to include the contour spacing as part of the inscription in each panel.

The inlets (or inscription blocks) of Figs. 2-4 and 6-8 provide parameter values of the Northern and Southern hemisphere with 1-2 digits, while the ratio of the value is given with three digits. This allows some space for speculations about the correct numbers as, e.g., for the upper left panel of Fig.7 with values for W_N and W_S between 6.4 and 7.6 GW. I think it would be better to provide about the same number of digits for the parameter values as for their ratio.

- This was an oversight on our part, and we agree with the reviewer's suggestion. We have revised Figures 2–4 and 6–8 accordingly.

Line 197: just below eq. (15) after "with d1 and d2" I miss a verb or "as"
Line 438: parenthesis for the reference not needed here
Line 461: "Bz" is probably meant here instead of "By"
Line 585: one "in" should be deleted
Line 618: "HH" is probably "HT"(?)

- These typos have all been corrected as suggested in the revised manuscript.

**Response to Reviewer #3 (Octav Marghitu)**

RC3: 'Comment on egusphere-2023-2920', Octav Marghitu, 17 Jan 2024

We thank the reviewer for their careful reading of our manuscript and their comments and suggestions. Below the reviewer's comments are shown in **black**, and our responses are shown in **blue**.

The paper Does high-latitude ionospheric electrodynamics exhibit hemispheric mirror symmetry? provides a comprehensive perspective over high-latitude ionospheric electrodynamics based on almost 9 years of Swarm data. The paper brings a significant contribution to the field and is certainly suitable for publication. Before that, however, perhaps there is room to optimize the transmission of the paper's message to the reader.

The (very) comprehensive character of the paper is both a merit and also an issue. Its substance could easily fill three papers, I would say, focused as follows and, piecewise, easier to absorb:

• One paper on the Hi-C convection model, perhaps including more details on the math (Section 3) in a less dense presentation;

• Another paper on combining Hi-C with AMPS to produce Swipe, and the resulting maps on electromagnetic work and conductance, $W$, $\Sigma_P$, $\Sigma_H$. This paper could also benefit from more discussion of the results.

• The (a)symmetry between northern and southern hemisphere could be the object of yet another paper, once again assisted possibly by more discussion.

As of now, the (a)symmetry 'paper' also gives the title of the full manuscript, while the visibility of the other two 'papers' is somewhat obscured. This might be regarded as a weakness, even if a rather uncommon one.

- We agree with the reviewer that this study is outside the norm in terms of length (also indirectly hinted at by the second reviewer), and we agree that our choosing to squeeze all three "papers" into a single study is not unproblematic. For us, this choice came down to a simple question of available bandwidth; we found it most expedient to write a single and perhaps unfortunately lengthy paper.

  We have attempted to partially make up for this weakness by making the model visible (announced so far on three different mailing lists), open source, and easily accessible (available on PyPI, GitHub, Swarm VirES, and Zenodo).

In the following I make a few comments on each of these three 'papers', then list a few more issues that may require the authors' attention. More comments on Hi-C, which feels also the more demanding.

1. The Hi-C 'paper'

a) The derivation of 2D maps from the 1D cross-track TII measurements may deserve some discussion beyond the math. To some extent, this has been done also before, by Lomidze et al. (2019), and was validated by comparison with Weimer (2005). However, Lomidze et al. (2019) concentrated on the cross-track component, whereas here both components of the convection are derived. My understanding

is that the 'jump' from 1D individual measurements to 2D statistical results is related essentially to the potential nature of the electric field, that prevails most of the time, and then convection is dominated by electric drift. But I wonder if the 1D character of the measurement does not still have some impact, in particular on the accuracy / error margin of the results (see also 4e). While the error margin is beyond the scope of the paper (see also point 3), it may still be worth to comment on this matter. For example, when the convection map shows plasma velocity mainly in E-W, cross-track direction, I expect this result is more accurate than plasma velocity shown by convection map mainly in N-S, along-track direction. To some extent, this reminds me the SuperDARN maps, where the model is based on solar wind parameters, though one expects better accuracy around the radar measurement points.

- We confirm the reviewer's understanding that the jump from 1D cross-track measurements to a derived 2D convection pattern relies heavily on the (assumed) potential nature of the electric field. We completely agree with the reviewer that this is an issue; it is the reason for the statement made on Line 255 of the original manuscript, where we state that "model convection in the north-south direction is … essentially unconstrained by measurements."

  In the revised manuscript we have inserted the following new paragraph on Lines 528–541:

  > Regarding the suitability of Swarm TII measurements for development of an empirical ionospheric convection model, Lomidze et al. (2019) and Lomidze et al. (2021) showed that Swarm TII cross-track measurements and corresponding model outputs from the Weimer (2005) empirical model are very similar in a climatological sense. On the other hand Lomidze et al. (2021) found that the along-track drifts measured by vertical and horizontal TII sensors in general do not agree, and are significantly different from the along-track drifts predicted by the Weimer (2005) model which, among other things, clearly show anti-sunward flow across the polar cap. They concluded, "Overall, the results for the TII along-track indicate that some large-scale features in that component of ion convection cannot be captured by the current version of the Swarm along-track drift measurements, and data from the [vertical and horizontal] sensors can be different." For these reasons we have chosen to exclude along-track drift measurements in deriving the Swarm Hi-C model. The predicted along-track component of ionospheric convection therefore relies heavily on our assuming a potential electric field (Equation 21) and by imposing the constraint discussed in Section 3.2, and it should here be emphasized that we have not explicitly validated Swarm Hi-C model predictions of along-track convection. Since the along-track component is not constrained by measurements (see Section 3) the uncertainty of the along-track convection predicted by Swarm Hi-C must of necessity be larger. A more complete discussion of issues with along-track ion drift measurements is given by Lomidze et al. (2021).

b) The Modified Apex coordinates, MA-110, appear to play an important role in the formalism, taking care, e.g., of the mapping from Swarm altitude to ionosphere (L180–181, L221–227) or the distortions of the magnetic field (L193–194, L575–576). For details regarding MA-110 the reader is referred to the paper by Richmond (1995). While this is in principle fine, it would be good to provide more clarifications, starting with the definition of the apex altitude (L202). Other features that could be detailed a bit are the way MA-110 takes care of i) mapping and ii) magnetic field distortions (mentioned above), iii) the possible bias of the SH model (L190–194), iv) how strongly non- orthogonal are ($d_1$, $d_2$ – L197) and ($e_1$, $e_2$ – L207)

(both pointing roughly in the same directions), v) what is roughly the difference between the two and the standard spherical system (L223–224) (the energy argument at L237 suggests perhaps less than 1°?), vi) some brief explanation of the difference between 44° QD latitude (L166) and 47° MA-110 (L250, L256) (is 47° MA-110 just the average of 44° and 50° QD, L251? how significant is actually the difference between QD and MA-110?).

- We thank the reviewer for pointing out that many of these points simply were not explained in enough detail. In the revised manuscript on lines 213–218 we now explain that the representation of the electric field and convection coefficients in Equations 16 and 18, respectively, explicitly indicate that these coefficients are constant along field lines, regardless of whether the field lines themselves depart from dipolarity (we know that they do). On these lines we have also made some general comments about how strongly nonorthogonal the various basis vectors are (poleward of ±60° MA-110 latitude, the angles between $\mathbf{d}_1$ and $\mathbf{d}_2$ and $\mathbf{e}_1$ and $\mathbf{e}_2$ do not deviate from orthogonality by more than 15° in either hemisphere), and referred the reader to the papers by Richmond (1995), Emmert et al (2010), and Laundal and Richmond (2016), who discuss many of these and related questions.

  Regarding the difference between QD and MA-110 coordinates, QD coordinates are not strictly constant along field lines, but MA coordinates (including MA-110 coordinates) are. So QD coordinates are good for things that vary with height, such as the B-field of divergence-free currents, while MA coordinates are good for things that map along field lines, like E-fields, convection velocities, and FACs.

  Laundal and Richmond (2016) also describe that in practice, the difference is essentially the reference height to which the dipole mapping is done. In QD coordinates, the reference height is the height of the point in question. In MA-110 coordinates, the reference height is 110 km. This means that if the point in question is at $h=110$ km, QD coordinates and MA-110 coordinates are identical. For points above this height, the QD latitude is equatorward of the MA-110 latitude, and vice versa for points below this height. (See Equation 43 in Laundal and Richmond, 2016). At Swarm altitudes of ~400–500 km, an MA-110 latitude of 47° corresponds to QD latitudes of ~45.4°–45.8°. These points are discussed on lines 542–547 on the revised manuscript.

c) The decimation at L168-169 is not explained (together with the related limitation to scales larger than some 40 km). Is this because of further matching with AMPS?

- On lines 170–172 of the revised manuscript we point out that this choice was made because we found that increasing the effective measurement cadence (i.e., including more measurements) did not visibly affect the shape of the potential patterns.

d) The two paras at the end of Section 6, L556-569, are quite helpful to understand the math, but they might fit better to Section 3, where the math is done (perhaps in Section 3.2?). Those paras are quite specific, they do not seem to belong to the Discussion section.

- This is a fair point, in the revised manuscript we have moved these paragraphs to Section 3.4 on lines 337–349.

2. The Swipe 'paper'

a) I think an important issue here is the error margin. While the authors eliminate problematic regions by asking the Hall and Pedersen conductance to be positive (with more constraint on Pedersen, via the threshold of 0.5 mW/m^2 in Eq. 37), the error margin is deferred to another study (L383–386). Nonetheless, a rough estimate of the uncertainty of $W_N$ and $W_S$ is provided at L436-437.  Could this estimate be briefly explained? And perhaps similar estimates could be provided and tentatively discussed for other quantities, like the conductances? Based on 'common sense' knowledge, conductances around a couple of mho and less are more and more uncertain, the lower the conductance is. On the other hand, low conductance areas are actually quite broad and can make a significant contribution to Joule heating, which is a major driver of ionospheric electrodynamics.  More discussion on the error margin seems appropriate, even if the actual (quantitative) solution is beyond the scope.

- We agree that the question of uncertainty is one that deserves serious attention, and as the reviewer notes, we think this is most properly addressed as part of a dedicated study. To our knowledge, no existing study involving development and demonstration of an empirical model of quantities such as ionospheric potential has yet taken up the question of uncertainty estimation.

  We agree that the statement that we made in the original manuscript was too vague to be of much use. On line 465 of the revised manuscript, we now refer the reader to Appendix B where we state how one may use standard error propagation techniques together order-of-magnitude estimates of the uncertainties of the magnitudes of J, E, and the angle between them to arrive at a rough estimate uncertainty of the hemispheric integrated EM work that turns out to be of order a few GW.

  Regarding the uncertainty of the conductance: Since the conductance depends critically on having a proper estimate of the neutral winds and is generally much more poorly understood than the electromagnetic work and/or Joule heating, we feel it is best to defer discussion of the uncertainty of conductance estimates to a dedicated study. We hope the reviewer will understand that our desire is to do the subject justice, and we do not feel this is possible within the present manuscript given the (already very large) scope of the study.

b) Related to this matter, the threshold of 0.5 mW/m^2 for W is explained at L379–382 based on typical values of the constituents – neutral velocity, magnetic field, and sheet current – together with a qualitative remark on apparition of very sharp gradients in the conductance below this threshold.  Given the importance of this threshold for the low conductance areas and for the validation of the results (L138–139), it would be nice to elaborate a bit and quantify roughly the very sharp gradients.

- Rather than adding more text, we have added a note on lines 410–411 of the revised manuscript where we state that we include a Supporting Information figure that shows typical distributions of the conductances when no screening criteria are applied. This is the figure:

[Figure]

Here one can see, for example, locations indicated by black boxes where the conductances change by as much as 50 mho from one grid cell to the next. There are also many cells within which the conductance is negative.

Of course, the locations of the sharp gradients and negative conductances are not static, but vary with IMF orientation, solar wind speed, and such. We have not spent any significant amount of time trying to identify where the negative conductances or sharp gradients occur, since we know that our model is missing key information about the neutral winds.

3. The (a)symmetry 'paper'

a) In the literature one can find two different perspectives on the (a)symmetry between northern and southern hemispheres: Papers of the sort here, that look at the northern and southern hemisphere under similar conditions of tilt angle and IMF By, as well as papers that concentrate on the instantaneous asymmetry – driven, to a large extent, by the different conductance between the summer and winter hemisphere (and also by the tilt angle and By, whose values are not mirrored for such studies). Judging just by title, one could question what is the perspective here, before the matter becomes clear in the text. It may still be worth to comment a bit on these two complementary facets of the (a)symmetry.

- We appreciate the reviewer's making this distinction between the types of studies that one encounters in the literature. In the revised manuscript we have made the following addition to the first paragraph of the discussion on lines 518–522, which relies heavily on the language that the reviewer has used (we thank the reviewer and have acknowledged their contribution in the acknowledgements):

    In the literature one encounters different approaches to the topic of symmetry between the two hemispheres that may be roughly separated into two categories: those that examine

asymmetries in the NH and SH under complementary conditions of tilt angle and IMF By, and those that concentrate on instantaneous asymmetries that are driven, to a large extent, by differences in conductance between the summer and winter hemispheres (but also by tilt angle $\psi$ and IMF $B_y$, whose values are not mirrored). This study belongs to the former category.

4. Other issues

a) L153–154: This sentence is correct, but the association between the neutral wind and the definition of the Poynting flux (Eq. 14), rooted in the energy conservation Poynting theorem, may drive some confusion.

- Good point, in the revised manuscript we have made the following change:

   The neutral wind $\mathbf{v}_n$ notably does not appear in Equation 14, as the Poynting flux is frame dependent and arises in connection with the well-known energy conservation (Poynting) theorem.

b) L155: Perhaps Methodology and data for Hi-C? This is, indeed, the core, but methodology includes also combination with AMPS, and then exploring (a)symmetry in all quantities (the three 'papers'…).

- In the revised manuscript we have changed the name of this section to "Methodology and data for Swarm Hi-C model" as suggested.

c) L156 – 158: Perhaps say simply that \hat{y} = \hat{x} \times \hat{r} / |\hat{x} \times \hat{r}|?

- Thank you for the suggestion, we have revised the manuscript accordingly.

d) Eq. 22: The tilt angle is not under sin or cos, like the clock angle. Is this because the tilt angle is (rather) small?

- Precisely, this is because the tilt angle remains within a (relatively) narrow range of ~±30°.

e) L304–305: Please explain briefly the origin of the ill-condition. Can this be related, at least to some extent, to the 1D measurements (point 2a above)?

- As acknowledged above, the fact that the measurements are 1D must have some influence on the derivation of the model. However, it is unlikely that the need for regularization is connected with the fact that the measurements are 1D. Without regularization the cost function would be entirely dependent on measurement-model misfit (first term on RHS of Equation 35). With such a simple cost function the norm of the model parameter vector $\mathbf{k}$ generally ends up being far too large, corresponding to overfitting or in the worst case numeric overflow.

f) L312–313: Not sure I understand: sectorial resolution, associated with M, is not the same with zonal (or longitudinal)? And N is not associated with latitudinal resolution, rather than zonal?

- We apologize for the possibly confusing choice of words here. On lines 321–322 of the revised manuscript we now write "That $M$ is much less than $N$ indicates that the longitudinal resolution of the model is much lower than than the latitudinal resolution."

g) L372: Could you describe / illustrate briefly the differences between Hi-C and AMPS?

- The difference between the models that we had in mind, which is implicitly referred to here but not explicitly stated, is that the perpendicular current vector **J** from AMPS and the electric field vector **E** from Swarm Hi-C are not in any way "co"-constrained to ensure that $\mathbf{J} \cdot \mathbf{E} > 0$ and $(\mathbf{J} \times \mathbf{E}) \cdot \mathbf{b} > 0$, as required by the physics (NOTE: assuming the neutral wind is zero in the Earth's rotating frame of reference!). In the absence of information about the neutral wind, it seems impossible to make a meaningful comment about how the models might be different if they had been co-constrained.

h) Eqs. 37 indicate a difference in the treatment of $\Sigma_P$ and $\Sigma_H$, in that for $\Sigma_P$ some margin is considered above zero (via the 0.5 mW/m 2 threshold). Please comment a bit on this difference, how comes that no margin is needed for $\Sigma_H$ ?

- The reason for imposing a different threshold is that we found the two thresholds correspond to different types of issues with the conductance distributions, as discussed on Lines 376–378: the 0.5 mW/m² criterion primarily addresses the issue of negative or unphysically large conductances or large conductance gradients within the polar cap and equatorward of ±60° MLat, while the $\Sigma_H >$ 0 mho requirement mostly addresses locations poleward of 70° where the Hall conductances may be negative (between -6 and -1 mho for the conditions we examined).

  In other words, these values were chosen on a heuristic basis, and we by no means intend to imply that they are somehow the "right" values (please see our comment to the second reviewer on this point). The simple fact of the matter is that for the conditions we examined, when we reduced the first threshold to 0 mW/m² or even 0.3 mW/m², we found that the Pedersen and Hall conductance distributions still evinced sharp gradients and/or negative or extremely large values. We chose the (approximate) minimum value necessary to mask regions with these problems, and found that this minimum value corresponds to what might be considered the typical contribution from a height-averaged neutral wind with a magnitude of 100 m/s and a current sheet density of 100 mA/m, as described on lines 379–382.

  We have added some explanatory remarks along these lines to lines 591–598 in the revised manuscript.

i) L392–394: This appears to hold in particular for local winter.

- Thank you for your careful examination of these figures, have added this comment on the relevant line (423) in the revised manuscript.

j) L414–415: neutral wind field corotating with the Earth is a bit confusing. Strictly, this means no neutral wind, whereas what is likely meant is that the neutral wind has the same direction as the Earth rotation.

- This is of course a question of frame of reference, such that if one is standing on Earth we do indeed mean $\mathbf{v}_n = 0$ (no neutral wind), whereas if one is looking down at the Earth in, say, a GSM frame of reference the neutral wind field rotates together with Earth.

  We thank the referee for pointing out that this may be a source of confusion, and on lines 444–445 of the revised manuscript we have changed the phrase "reducing the electric field in the neutral wind frame" so that it reads "reducing the electric field in Earth's rotating frame of reference".

k) L423–424: Any comment on the dominant direction of the neutral wind? (considering also the previous para)

- We agree that this would be valuable to comment upon. Unfortunately we do not feel comfortable commenting on the dominant direction of the neutral wind, as the presence of neutral wind shears at the relevant altitudes greatly complicate determination of a dominant direction. As mentioned on Line 125, this is a topic that we are actively researching.

l) L451 and L480: The Supporting Information is missing (Figures S1–S6).

- Thank you (and the other reviewers) for catching this oversight on our part. We have included them as part of our resubmission.

m) L484–486: This suggests that hot spots might be related to IMF By (?).

- This is a good point that we have included on line 516 of the revised manuscript

5. Typos and alike

L101 and L129: Define the LHS and RHS acronyms;

L181 and Fig. 1: The black lines are hardly visible, change black to some color (?). Add scales to the distributions (?);

- We have increased the thickness of the black lines. We also attempted to add scales to the distributions, but these affected the layout of the figure without my (SMH) managing to find out why. We hope the reviewer can understand that we ultimately decided to leave off the scales.

L208: Laundal et al. (2018) useD;

L224, L225: EquationS 16, 18;

L312: sectorial, lower than than;

L341: If the average CPCP is 51 kV for both NH and SH, it cannot be 3\% greater in the SH, as stated in the caption of Fig. 2;

- We appreciate the referee's attention to this detail. We find after recalculation that the average in the SH is 52.25 kV and the average in the NH is 51 kV, such that the SH average is 2% greater than the NH average

L355: similar -> more or less similar (?);

L362: contours -> colored contours (?);

Figs. 6–8: The black line and color contours can be compared easily in terms of shape, less easily in terms of value. The integral values at the top right corners help;

L409–410: Delete from 'as mentioned…' to the end of the sentence.

L439: Regardless of B T (?);

L460: By -> IMF By;

L461: IMF By -> IMF Bz;

L474: distribution OF;

L516: examines (?), mirror asymmetry -> mirror symmetry;

L522: These studies are of relevance to this study;

L537-538: Weimer and Edwards (2021) -> WE21;

L584: field field;

L618: HH -> HT (?).

- Thank you for catching all of these typos and small mistakes. We have corrected all of them in the revised manuscript, aside from where noted above.